# Endogenous learning for green hydrogen in a sector-coupled energy model for Europe

Elisabeth Zeyen [1,2] ✉, Marta Victoria [3,4] & Tom Brown [1,2]

Many studies have shown that hydrogen could play a large role in the energy transition for hard-to-electrify sectors, but previous modelling has not included the necessary features to assess its role. They have either left out important sectors of hydrogen demand, ignored the temporal variability in the system or neglected the dynamics of learning effects. We address these limitations and consider learning-by-doing for the full green hydrogen production chain with different climate targets in a detailed European sector-coupled model. Here, we show that in the next 10 years a faster scale-up of electrolysis and renewable capacities than envisaged by the EU in the REPowerEU Plan can be cost-optimal to reach the strictest +1.5°C target. This reduces the costs for hydrogen production to 1.26 €/kg by 2050. Hydrogen production switches from grey to green hydrogen, omitting the option of blue hydrogen. If electrolysis costs are modelled without dynamic learning-by-doing, then the electrolysis scale-up is significantly delayed, while total system costs are overestimated by up to 13% and the levelised cost of hydrogen is overestimated by 67%.

Today hydrogen plays a minor role in European final energy consumption with a 2% share[1], most of which is produced from natural gas with associated $CO_2$ emissions. In the future, however, hydrogen produced with low carbon emissions is likely to play an increasing role in the energy mix as decarbonisation progresses with shares of up to 24% of the final energy demand by 2050[2]. Applications could include the production of green steel[3] or synthetic fuels[4,5]. There are several ways in which hydrogen can be produced: (i) from fossil fuels such as coal or natural gas, (ii) from fossil fuels in combination with $CO_2$ capture, or (iii) via electrolysis. The production costs for hydrogen depend in the first two cases on the price of fossil fuels, and in the second case on the cost of electricity. In order to reduce dependence on fossil fuels with high gas prices and to accelerate the energy transition, the European Commission aims to boost the deployment of hydrogen electrolysis. This objective has been concretised in the REPowerEU plan[6] published in May 2022, which has set a goal to produce 10 million

tonnes of hydrogen with renewable electricity and to import further 10 million tonnes to the EU by 2030.

Hydrogen produced from water electrolysis using renewable electricity, so-called green hydrogen, is considered to be crucial to decarbonise hard-to-electrify sectors of the energy system. Since it is an immature technology, it is highly likely that costs and efficiencies will improve as production scales up, a phenomenon known as 'learning-by-doing'. Several studies have been carried out on how future hydrogen electrolysis investment costs develop over time or with capacity deployment. However, many are based either on predefined production or capacity developments[7–9] or on expert surveys[10]. Way et al.[8] explore various cost distributions for the world's energy system up to year 2070 for multiple technologies via a Monte Carlo approach. Through the different investment cost trajectories they cover uncertainties in the cost developments, but the path for installed capacities are exogenously defined and energy dispatch is not modelled.

[1]Department of Digital Transformation in Energy Systems, Faculty of Process Engineering, TU Berlin, Einsteinufer 25 (TA 8), Berlin 10587 Berlin, Germany. [2]Institute for Automation and Applied Informatics (IAI), Karlsruhe Institute of Technology (KIT), Forschungszentrum 449, Eggenstein-Leopoldshafen 76344 Baden-Württemberg, Germany. [3]Department of Mechanical and Production Engineering, Aarhus University, Inge Lehmanns Gade 10, Aarhus 8000, Denmark. [4]Novo Nordisk Foundation CO2 Research Center, Gustav Wieds Vej 10, Aarhus 8000, Denmark. ✉e-mail: e.zeyen@tu-berlin.de

The concept of the experience curve, which means that costs and efficiencies improve as production increases, is not a new idea. Already in 1936, Wright[11] described cost reductions related to aircraft production in a mathematical unit cost model. Since then, global cumulative production volume or overall installed capacity is used to quantify experience, attaining a good match with reality in many different technologies. However, experience curves are often neglected in energy system models, since they result in non-linear, non-convex optimisation problems which adds complexity. Modelling the endogenous cost curves leads to a better representation of the dynamics of new technologies[12], but requires a significantly higher computational effort. This is especially important when modelling with perfect foresight over a large time horizon. If the cost and efficiency improvements are given exogenously to the model, the model can 'wait' with investments until they are profitable, while in reality, costs would only dynamically decrease as investments take place. Nevertheless, in many models, technology trends including cost reduction or efficiency improvement, are only considered as exogenously assumptions. An example of how these exogenous assumptions overestimate future investment costs is solar photovoltaic (PV). The investment costs of solar have decreased rapidly due to successful policy support. A comparison by Krey et al.[13] of the cost assumptions in 15 Integrated Assessment Models (IAM) shows that the cost of PV in 2020 have already fallen below the model expectations for 2050. Way et al.[8] show that the progressive cost projections for solar, onshore wind, batteries and polymer electrolyte membrane (PEM) electrolysis from several IAMs and the International Energy Agency (IEA) are high compared to historical developments or even above costs in 2020. Several other studies[14–17] criticise that the link between cost reduction and capacity installation is not well represented in models and that exogenously-set constraints such as floor costs or excessively low annual growth rates lead to an underestimation of cost reductions. This highlights the need to model investment costs endogenously without assuming unduly elevated floor costs, overly limiting constraints on maximum annual expansion rates or maximum penetration of renewable energies.

Endogenous cost reductions are currently used in some IAMs[18–21] which consider learning in multiple sectors and global developments. But because of the low temporal resolution in these models, they cannot represent the challenges of an energy system with a high share of variable renewable generation. In addition, most IAMs apply the Hotelling rule[22] to determine the $CO_2$ prices, which with endogenous learning is not always applicable[23]. Bottom-up techno-economical Energy System Models (ESM) can better capture the temporal variability of the energy system. However, studies applying endogenous learning only deal with the power sector[24–28] or focus on a single country[29].

In this work, we explore how learning-by-doing on the full hydrogen production chain interacts with a fully sector-coupled energy model. We apply endogenous learning for electrolysis and renewable energy in the European model PyPSA-Eur-Sec[30] with full sector coupling and enough temporal resolution to capture renewable variability. The period between 2020-2050 is investigated with seven investment periods and perfect foresight over the whole modelling horizon. We do not specify the maximum annual expansion rates of renewable capacities nor $CO_2$ emission targets for every single year. These additional conditions often lead to a lower computational effort but also predetermine the transition paths. In this way, we determine the cost-optimal annual expansion rates without making any assumptions about expansion limits. Such limits have been estimated to be artificially low in many studies compared to actual capacity developments. We want to address the following two research questions: What are the possible cost developments of green hydrogen production in Europe under the assumption of different $CO_2$ budgets without fixed capacity deployment projections? How do different methods of modelling cost reduction influence the results?

We consider three different competing options for producing hydrogen: (i) grey hydrogen (via steam methane reforming (SMR)), (ii) blue hydrogen (SMR + carbon capture, capture rate 90%), and (iii) green hydrogen (via electrolysis). Hydrogen can be used for methanation, for heating (hydrogen boilers), electricity (fuel cells and retrofitted open cyclic gas turbines (OCGT)), in the industry, and in the transport sector. The synthetic gas from methanation can be used in the heating sector (gas boilers or combined heat and power plants (CHPs)), for industry processes or in the power sector (OCGTs or closed cyclic gas turbines CCGTs). The demand pathway for hydrogen in parts of industry and transport is exogenously defined, while in all others sectors, hydrogen competes with other ways of supplying demand. This means both demand for hydrogen (e.g. where it competes with heat pumps in the heating sector) and the supply side (e.g. if the hydrogen is produced via electrolysis or SMR with or without carbon capture), are part of the optimisation. OCGT and gas boilers for heating can be retrofitted to run flexibly with natural gas or hydrogen (see Fig. 1).

There are various technologies for hydrogen electrolysis. In this study, cost and efficiency assumptions of alkaline electrolysis cells (AEC) are used since they are currently the most common electrolysers available on the market. We only consider large plants (above 100 MW) to avoid the scaling effects of very small plants. These costs consist of the equipment and installation costs. They include the expenses for the stack, power electronics, gas conditioning, balance of the plant and labour. The annual operational and maintenance costs are estimated in ref. 31 based on current projects to be 2% of the investment costs. The cost of replacing the stack is not included in the fixed operational and maintenance cost (FOM) since it is assumed that the stack does not need to be replaced within the technical lifetime. The investment costs do not include costs for water purification, transformer costs, or connection fees to the transmission system. AEC is currently the dominant technology, but there are other types of electrolysis, such as PEM or solid oxide electrolysis cell (SOEC), which may play a more prominent role in the future. We analyse the impact of higher initial investment costs (comparable to the investment cost of PEM) in the Supplementary Material. All costs are given in real 2015 Euro.

We minimise total system costs for the period 2020-2050 with perfect foresight and investments in new capacity in a five-years interval for three different $CO_2$ budgets. The investment costs of renewable generation capacity (solar, onshore and offshore wind) and hydrogen electrolysis are endogenous and thus a result of the optimisation in the form of a piecewise-linearised one-factor experience curve. This results in a mixed integer linear problem MILP which is further described in the Methods Section. Endogenous global learning of the renewable generation capacities for solar PV, onshore and offshore wind is applied. Global learning assumes that global capacity grows proportionally to European capacity. The relative factor corresponds to today's share. For example, if 1 GW of solar PV is newly built in Europe and the share of European solar PV capacity is 22%, the global capacity grows by about 4.5 GW. Further details about the initial investment cost assumptions of renewable generation are given in the Supplementary Material. For hydrogen electrolysis local learning is applied, since it cannot be assumed that global expansion will keep up with Europe. This means investment costs reduce only based on installed capacities within Europe. Learning rates are varied by ±10% for all learning technologies to understand the robustness of the results. Only the extreme cases are covered, i.e. a very optimistic scenario in which all default learning rates are higher by +10% and a pessimistic case in which all learning rates are -10% compared to the base case. The cost and efficiency assumptions for all other technologies are exogenously given depending on their build year.

In this study, we investigate the potential cost developments of green hydrogen production in Europe under different $CO_2$ budgets and analyse the impact of various cost reduction modelling methods.

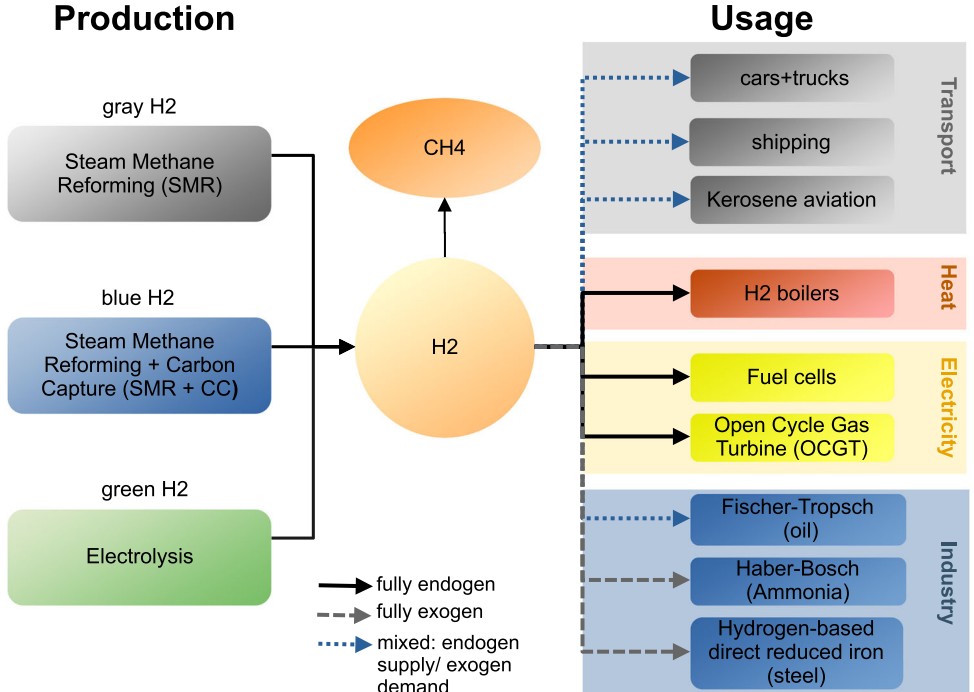

**Fig. 1 | Overview of hydrogen usage and production in the model.** In most sectors, both the production and demand of hydrogen is endogenous. With the exception of the production of ammonia and steel in industry, for these sectors fixed hydrogen demands are specified for the respective reference year. For the transport sector, e.g. the share of internal combustion engine (ICE) vehicles is predefined for each year (exogenous demand), and whether the fuel is of fossil or synthetic origin is optimised (endogenous supply).

Here, we show that hydrogen production costs drop to as low as 1.26 €/kg$_{H_2}$ by 2050. The production of hydrogen shifts from grey to green, bypassing the need for blue hydrogen. The scaling-up of electrolysis is delayed, and overall system costs are overestimated by up to 13%. The levelised cost of hydrogen is inflated by 67% if electrolysis costs are not modelled with endogenous learning-by-doing.

## Results
### Total system costs to achieve the carbon targets
To stay within the +1.5°C budget, the direct total annualised system costs (ignoring any costs of climate change damages) are up to 27% higher than the +2.0°C budget (Fig. 2). Cost are higher in the +1.5°C scenario because (i) existing assets with high $CO_2$ emissions are phased out before the end of their lifetime, (ii) large-scale investments are made before the costs are reduced by learning, and (iii) major parts of the oil demand are already supplied by synthetic fuels in 2030, which are more expensive to produce. However, total annualised system costs of the scenarios in 2050 are similar and vary only by 2% between the +1.5°C and +2.0°C budget since the investment costs decline. The annualised system costs of the +1.7°C scenario in 2050 are 2% lower compared to the +2.0°C scenarios. The reason is that investments in low-carbon infrastructure with the +1.7°C budget happen earlier, while for the +2.0°C scenarios a major infrastructure transformation with associated higher costs is needed in 2050 to meet the condition of climate neutrality. The European Union target of 55% greenhouse gas reduction in 2030 compared to 1990[32], applied to $CO_2$, is within the +1.7°C and +2.0°C scenarios (Fig. 3). If estimated costs of climate change damage with a social cost of carbon (SCC) of 120 € per tonne $CO_2$ are added, the total system costs are even slightly higher in the +2.0°C scenarios compared to the +1.5°C and +1.7°C scenario in the period 2025–2045 due to the higher $CO_2$ emissions (see Fig. S26 in the Supplementary Material). A previous study[33] deals in more detail with the impact of SCC on the total system costs.

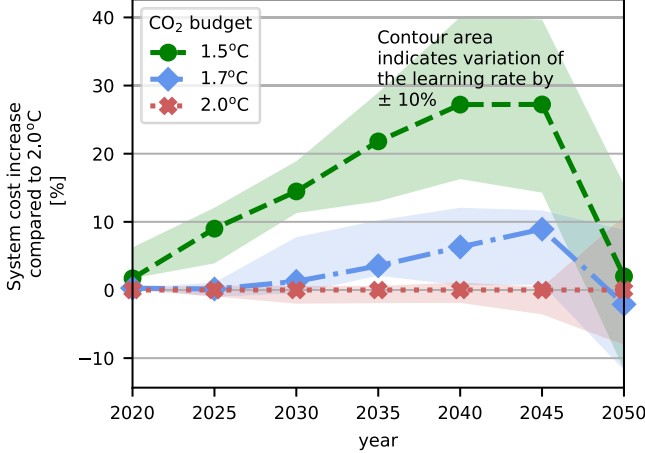

**Fig. 2 | Total annualised system costs compared to the endogenous +2°C budget scenario without estimated costs of climate change damage.** The costs of individual years are shown here, however, the entire period from 2020–2050 is solved in one single optimisation. The shaded areas represent the sensitivity analysis to ±10% learning rates for different technologies.

### Electrolysis investment costs
The investment costs of electrolysis decline strongly for all three assumed $CO_2$ budgets. For the base learning rate assumptions (Table 1), investment costs are for every year, and the budget is below the cost estimates of the Danish Energy Agency (DEA) (Fig. 4). The tighter the budget, the more renewable generation and electrolysis capacities are employed and the stronger is the cost decline. All scenarios lead to a significant reduction in the investment costs of electrolysis, ranging from 140 to 380 €/kW$_{elec}$ already in 2030. The largest cost reductions for the +1.5°C scenario occur until 2025, for the +1.7°C budget until 2030 while the investment costs in the +2.0°C scenarios

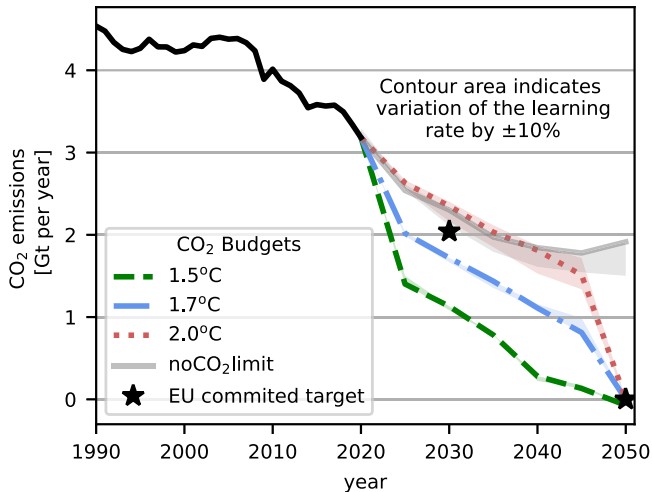

**Fig. 3 | Annual CO$_2$ emissions.** Three different budgets are assumed for Europe deducted from the global budget assuming equal per capita share (i) budget 25.7 Gt (+1.5°C), (ii) 45 Gt (+1.7°C), (iii) 73.9 Gt (+2°C). Further, carbon neutrality is required in all scenarios by 2050. The hashed line shows a scenario with the default learning rate presented in Table 1. Contour indicates scenarios with ± 10% variation of the learning rate for all technologies with endogenous learning. We compare the budget scenarios with a fourth scenario (noCO2limit) in which we do not specify any CO$_2$ constraints.

**Table 1 | Global capacity, European share of global capacity and learning rates (cost reduction for every doubling of cumulative capacity)**

| Technology | Global capacity [GW] | European share [%] | Learning rate [%] |
|---|---|---|---|
| Solar PV | 707[58] | 22[58] | 24[31] |
| Onshore wind | 699[58] | 26[58] | 10[31] |
| Offshore wind | 34[58] | 73[58] | 10[31] |
| Electrolysis | 1[159] | Local learning | 16[9] |

[1]only considering alkaline electrolysis and no other electrolysis type.

decline most strongly in later years between 2035 and 2040. In 2050, investment costs for electrolysis decrease down to 75 and 95 €/kW$_{elec}$ (compared to DEA 250 €/kW$_{elec}$) for +1.5°C and +2°C respectively (Fig. 4).

To examine the robustness of the results, the learning rates are varied by ± 10% in additional scenarios. In the scenarios with the most pessimistic assumptions (all learning rates are 10% less than the base case), the electrolysis investment costs range between 317 and 360 €/kW$_{elec}$ in 2050.

### Installed electrolysis capacities and hydrogen usage

The +1.5°C scenario sees the cost-optimal deployment of 435 GW of electrolysis which produce 36 million tonnes of hydrogen in 2030. This is clearly not realistic given the necessary scale-up of production facilities that would be required. In the scenario corresponding to a temperature increase of +1.7°C, 60 GW of electrolysis are installed by 2030 which produce 4 million tonnes of hydrogen. Under the +2°C scenario, 4 GW of electrolysis are deployed and 0.1 million tonnes of hydrogen are produced (see Fig. 5). The European Commission's REPowerEU plan[6] targets to produce 10 million tonnes of hydrogen within Europe, which corresponds to electrolysis capacities of 140 GW$_{el}$ assuming average utilisation factors of 43% and 70% conversion efficiency[34], and importing further 10 million tonnes of hydrogen from neighbouring countries by 2030. This target is in between our +1.5°C budget and +1.7°C budget result, given that we do not consider hydrogen imports from outside Europe.

The production of hydrogen switches from SMR (grey hydrogen) to electrolysis (green hydrogen) in all scenarios, while SMR in combination with carbon capture (blue hydrogen) is not installed in any scenario. To further explore the option of SMR with carbon capture, we run sensitivity scenarios for blue hydrogen production depending on capture rate, investment costs and available CO$_2$ sequestration potential (see Supplementary Material). Blue hydrogen is only produced at scale under certain optimistic conditions. For example, with our base assumptions for CO$_2$ storage potential (200 Mt$_{CO_2}$/a) and capture rate (90%), blue hydrogen has a share of 8% in total production with SMR investment costs of 286 EUR/kW$_{H_2}$ (50% of our reference assumption). A large sequestration potential of 2000 Mt$_{CO_2}$ per year

with base assumptions on investment costs (572 EUR/kW$_{H_2}$) and capture rate (90%) leads to production of blue hydrogen with a share of 19% of total production. With more optimistic assumptions on capture rate (100%) and CO$_2$ storage potential (2000 Mt$_{CO_2}$/a), most of the hydrogen is produced as blue hydrogen from investment costs below 286 EUR/kW$_{H_2}$.

The timing of the transition from grey to green hydrogen production differs between the scenarios. In the scenario with a +1.5°C budget, green hydrogen is already produced in 2025, while in the scenario with a +2°C budget hydrogen is supplied by SMR until 2040–2045 (see Figure S30 in the Supplementary Material for a breakdown of hydrogen supply and usage). For all budgets, hydrogen demand increases from about 110 TWh$_{H_2}$/a in 2020 to up to 4000 TWh$_{H_2}$/a in 2050. In the +1.5°C scenario, the hydrogen is primarily used to produce synthetic fuels (1800 TWh$_{H_2}$ in 2050) and synthetic methane (700 TWh$_{H_2}$/560 TWh$_{CH_4}$ in 2050). The produced synthetic methane is mainly used in industry processes (424 TWh$_{CH_4}$) but also in OCGTs (135 TWh$_{CH_4}$). There is an option to convert the OCGTs to run on hydrogen, but this option is not used. A smaller fraction of the hydrogen is used in the industry (500 TWh$_{H_2}$ in 2050), to power fuel cells (440 TWh$_{H_2}$ in 2050) and for shipping (200 TWh$_{H_2}$ in 2050). The option of retrofitting existing natural gas boilers for operation with hydrogen is not exploited. The natural gas boilers are largely replaced by heat pumps until 2040.

Hydrogen is stored in salt caverns and the energy capacity ranges between 278–349 TWh from the +2.0°C to the +1.5°C scenario. These storage capacities are comparable to existing natural gas storage of 1075 TWh, once adjusted for the lower volumetric energy density of hydrogen, and below the European technical potential of 84.8 PWh$_{H_2}$[35]. No costs are assumed in the main results for the hydrogen transport. A sensitivity analysis of a scenario with a higher spatial resolution and corresponding costs for electricity and hydrogen infrastructure, as well as an annual breakdown of hydrogen supply and usage and installed capacities, is shown in the Supplementary Material. In scenarios with a higher spatial resolution, the hydrogen and electricity grids account for a share of 0.1-7.6% of the total system costs, higher electrolysis capacities are installed and total system cost increases by 12-16%. The trade-offs of an electricity and hydrogen grid in a decarbonised European energy system are discussed in more detail in a further publication[36].

### Renewable generation costs and capacities

In order to achieve net-zero emissions in 2050, renewable generation capacity must be strongly expanded in all scenarios to at least 3.2 TW solar, 1.7 TW onshore wind, and 175 GW offshore wind. The timing of the capacity build-out differs across the individual scenarios: with an ambitious budget, there is a strong expansion between 2030–2040, while in the other scenarios the strongest expansion takes place in later years from 2040 (+1.7°C) or 2045 (+2.0°C). The REPowerEU plan[6], with targeted PV capacities of 750 GW$_{DC}$[37] (which corresponds to 600 GW$_{AC}$) by 2030, lies within our results of +1.7°C with 764 GW and +2°C with 422 GW. However, the wind expansion targets of REPowerEU with

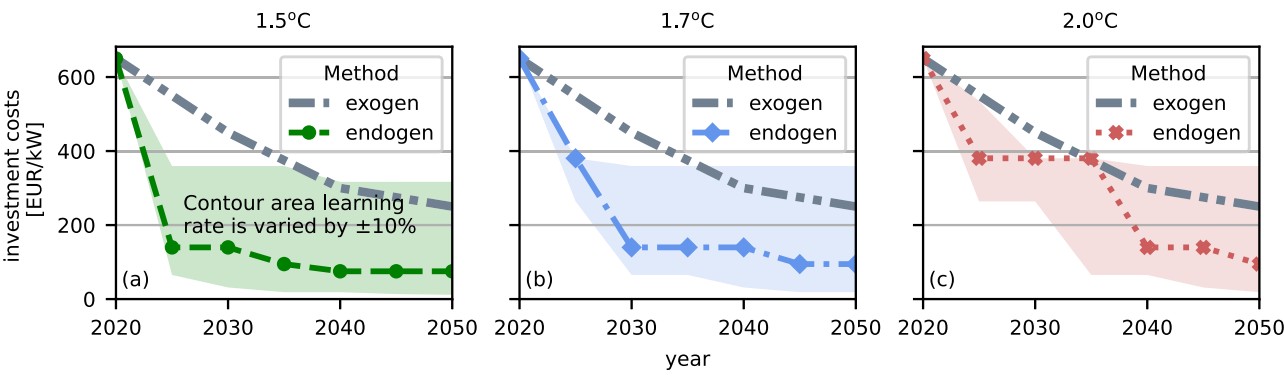

**Fig. 4 | Investment costs of electrolysis for different carbon budgets.** Hashed line shows a scenario with the default learning rate presented in Table 1. Contour indicates scenarios with ±10% variation of the learning rate for all technologies which undergo learning. Investment costs of the other endogenous learning technologies are shown in Supplementary Material Fig. S23.

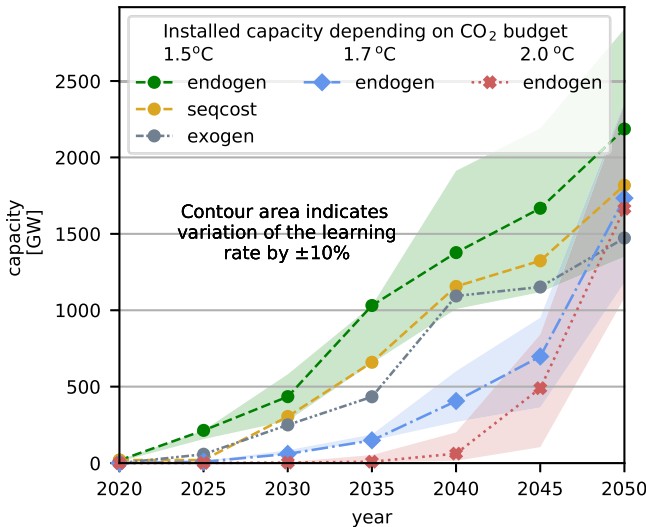

**Fig. 5 | Electrolysis capacity of the three different budgets.** For the +1.5°C scenarios also the three different methods are compared. The sequential cost (seqcost) and the exogenous (exogen) method results in later investments and underestimations of installed electrolysis capacities compared to the endogenous method (endogen).

480 GW[38] by 2030, lag behind our findings with at least 555 GW in the +2°C scenario by 2030.

In scenarios with base learning rate assumptions and endogenous learning, investment costs (without grid connection costs) for renewable generation capacity in 2050 range between 171–237 €/kW for solar PV, 818–900 €/kW for onshore wind, and 1327 €/kW for offshore wind from the tight +1.5°C to the +2°C budget (see Figure S23 in the Supplementary Material). These are below the DEA's cost projections. In particular, the costs for solar PV decrease significantly and are 43% lower in 2050 with ambitious climate targets than the DEA estimates of 300 €/kW. One should be aware that grid connection costs are added to these investment costs in the model that do not undergo any learning. For offshore wind, grid connection costs depend on the location and connection type, while for solar and onshore wind additional grid connection costs of 133 €/kW are added.

### +1.5°C budget hard to accomplish

The build-out rates of renewables and electrolysis in scenarios with a +1.5°C budget are significantly higher than the historical record, which emphasises the challenge to stay within this budget. For example, the

average annual build out rate is 77 GW for solar PV in the +1.5°C scenario between 2020–2030. This corresponds to almost a tripling of the historical maximum annual expansion rates in Europe[39]. The same applies to onshore and offshore wind, with average annual build-out rates of 97 GW and 16 GW between 2020–2030 in our scenario, which are roughly five times the historical maximum annual build-out rates[40,41]. Between 2030–2040 even higher build rates are required of up to 200 GW solar capacity per year while the annual expansion of onshore wind decreases to 58 GW. A sensitivity analysis, in which the maximum annual expansion rates of renewables are limited, can be found in the Supplementary Material. Limiting maximum annual expansion rates leads to higher costs because more offshore wind and nuclear power plants are deployed instead of less costly onshore wind and solar. Hydrogen production is lower because methanation is no longer favoured. The option of producing blue hydrogen is not used. One should be aware, that we do not assume any negative emissions after 2050, nor any additional sufficiency measures, nor do we allow a fully endogenous transition path for the transport sector. These are all factors that could reduce the necessary build-out rates and make the +1.5°C scenarios more achievable.

### Scenario without any carbon target

In one further scenario, no constraints on $CO_2$ emissions are assumed in order to investigate what a cost-optimal system without a carbon target looks like. As in the previous scenarios, the investment costs of renewables and green hydrogen are subject to endogenous experience curves. Compared to the current system, the share of renewable hydrogen increases even without climate targets. $CO_2$ emissions reduce from historical levels of 3.2 Gt/a in 2020 to 1.9 Gt/a in 2050 (Fig. 3). Emissions in 2030 are 2.3 Gt/a, which is only slightly above the EU's 2030 target. Hydrogen is largely produced via SMR. In 2050, 23% of the hydrogen demand in the $CO_2$-unconstrained scenario is covered by green hydrogen via electrolysis. $CO_2$ emissions stagnate from 2040 onward even with very high learning rates, whereby a large part of the emissions is generated by the use of fossil gas in the heating and power sector, as well as emissions from aviation and feedstocks for the petrochemical industry. It should be noted that our $CO_2$-unconstrained scenario does not continue historical trends, but allows for the transformation of energy sectors if it is cost-effective. For example, even without a set $CO_2$ limit, the comparatively expensive coal-fired power generation is phased out and ICE vehicles are replaced by electric vehicles and hydrogen-powered trucks, as in the other scenarios with a $CO_2$ budget. Sectors in which decarbonisation is not cost-optimal, are not transformed and continue to generate emissions, such as the production of feedstocks for the petrochemical industry.

## Comparing different methods of modelling technology learning

In this section, we examine the difference that modelling learning-by-doing dynamically makes to the results. We compare three different methods of modelling investment cost reduction typically used in ESM, including (i) the endogenous method where investment costs are adjusted according to the installed capacity, (ii) the standard exogenous method with given fixed cost trajectories for each technology and investment year and (iii) the sequential method. In the sequential method, the optimisation problem is first solved using the exogenous cost assumptions, then the investment costs are updated depending on the optimised installed capacities. The same experience curves are assumed as in the endogenous method. The process of solving and updating investment costs is iteratively repeated until the difference of investment costs between two iterations is below a threshold. The threshold is set to a maximal mean square difference between optimised investment costs of the current and the previous iteration of 5%. The sequential method is a linear problem that requires less computational resources than the MILP of the endogenous method, but unlike the exogenous method, the investment costs are adjusted based on the installed capacities.

The cost difference of not using endogenous learning-by-doing is shown in Fig. 6. In contrast to the exogenous method, the investment costs of the endogenous and sequential method depend on the installed capacities. For the +1.5°C budget, the sequential and exogenous methods result in up to 7% and 13% higher annualised total system costs compared to the endogenous method. The cost difference between the +1.5°C and the +2.0°C budget is smaller with the endogenous method compared to the sequential and exogenous method, because although larger capacities of renewables and electrolysis are needed at an earlier point in time, the costs also decrease more due to the faster scaling up and the foresight of the endogenous method of potential investment cost reductions. The endogenous method, therefore, allows a better comparison of costs of scenarios with different infrastructure needs or transition speeds.

The endogenous method leads to an earlier deployment of the learning technologies since it is the only one of the three methods that has the foresight of how far costs can decrease. As a result, investments are made early in order to reduce investment costs. With the sequential and exogenous methods, investment is delayed and the model 'waits' until costs fall. For example in +1.5°C scenarios in 2025, 214 GW of electrolysis are installed in the endogenous scenarios and only 20 GW electrolysis in both the sequential and exogenous cost scenarios. With the endogenous method, the hydrogen is used for the synthesis of fuels and methane earlier and to a larger extent. In scenarios corresponding to a +1.7°C budget about 1800 $TWh_{H_2}$ are produced in 2045 mainly via electrolysis, while only half of this amount is produced in the exogenous case. The larger volume of hydrogen is used in the endogenous case for synthetic fuels while in the exogenous case they have a fossil origin. This is particularly noteworthy since most ESM assume exogenous cost reductions and thereby underestimate initial investments in early years (Fig. S30 in the Supplementary Material). The foresight of the endogenous method is particularly important when modelling the dynamics of emerging technologies for which strong cost reductions are possible. The endogenous method can consider the potential cost development during optimisation and make investment decisions based on this. With the sequential and exogenous method, the investment decisions are strongly subject to the initial assumed cost projection.

The prices for hydrogen are an output of the linear optimisation and are determined by the dual variables. For the exogenous and sequential method, these are obtained directly from the optimisation. In the endogenous scenarios, the investment costs and capacities of the learning technologies determined from the optimisation are assumed fixed and the optimisation is rerun as a linear problem. The exogenous method overestimates the cost of hydrogen by up to 67% in 2030 and 38% in 2050 compared to the endogenous scenarios. For example, in endogenous scenarios with a +1.5°C budget hydrogen costs drop to 1.32 $€/kg_{H_2}$ in 2030, while in the sequential and exogenous scenarios they are 1.95 $€/kg_{H_2}$ and 2.22 $€/kg_{H_2}$ respectively. In 2050, the costs in the endogenous scenarios reach 1.26 $€/kg_{H_2}$ and the exogenous ones 1.73 $€/kg_{H_2}$.

The system composition in 2050 is primarily influenced by the assumed $CO_2$ budget and not by the modelling method (see Fig. 5). However, our results show deviations between the methods. The exogenous method leads to the lowest installed electrolysis capacities in 2050 followed by the sequential method with -33% and -17% respectively compared to the endogenous method in the +1.5°C scenarios. This can be explained by the lower investment costs for electrolysis obtained in the endogenous scenarios (75 $€/kW_{elec}$ in 2050 with a +1.5°C budget), which are below the sequential optimised investment costs of 79 $€/kW_{elec}$ and the exogenous assumptions of 250 $€/kW_{elec}$. The produced volume of hydrogen differs between the methods. For example, in scenarios corresponding to a +1.5°C budget, 24% and 30% less hydrogen is produced with the sequential and exogenous methods in 2030. With the endogenous method, more hydrogen is used for methanation and for re-electrification in fuel cells compared to the other two methods.

Technology learning depends on global developments while most ESM model single regions or continents. Cost in Europe, for example for solar PV, would reduce if solar panels are employed at a large scale in China. In this study, we have assumed that global renewable capacities increase in proportion to the growth in Europe. For electrolysis, we assume local learning. The exogenous method can provide a better description of cost developments for technologies for which large expansion rates outside the modelled region are expected. One could circumvent this problem with the sequential and endogenous method by splitting up the technology learning into two factors describing regional and global learning. The regional factor depends on the installed capacities in the regional model, while the global factor describes an exogenous cost decrease dependent on expected global development. The split into two factors was outside the scope of this study.

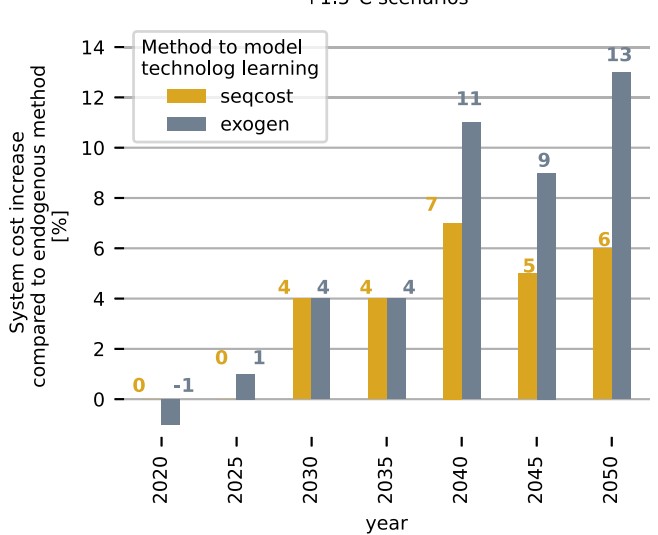

**Fig. 6 | Difference in total system costs for +1.5°C scenarios compared to the endogenous method with base learning rate.** Exogenous (exogen) and sequential (seqcost) method result in higher total annualised system costs compared to the endogenous scenarios. The comparison for the +1.7°C and +2.0°C scenarios and the impact of different learning rates can be found in the Supplementary Material S28.

The exogenous and sequential method require significantly less memory and computing time compared to the endogenous method. For example, using the commercial solver Gurobi[42] with 12 threads, the scenarios with endogenous learning need about 21 hours to solve and require 30 GB RAM, while the sequential models solve in less than 1 hour with only 3 GB RAM and the exogenous method solves in 1 minute with only 3 GB RAM. The sequential and the exogenous method require lower computational effort and thus offer the possibility to calculate e.g. with higher spatial resolution. The solution time and memory requirements for the endogenous method increase with the number of modelled learning technologies. In contrast, the computational effort of the sequential method is independent of the number of learning technologies, since the costs are only updated after optimisation. This allows us to consider cost reductions for all technologies depending on the installed capacity without increasing the computing time.

The chosen method of modelling technology learning should depend on the particular research question. The sequential and exogenous method are advantageous if the available computational resources are small or a greater level of detail (e.g. higher spatial resolution) is important. The number of technologies that are subject to learning can be increased with the sequential method without additional computational effort. The exogenous method can represent developments outside the modelled region. If the cost reductions are expected primarily through local learning within the modelled region, the sequential and endogenous methods are appropriate. The endogenous method is favourable if the timing of the investment is analysed, costs of different climate target scenarios are compared or trade-offs between emerging technologies are considered. Technologies with a large learning potential can be better modelled with the endogenous method, since small additional capacities already lead to strong cost reductions.

## Discussion

In the following, we compare our costs for green hydrogen production with other studies and outline the limitations of this work. In the Supplementary Material, we offer a more comprehensive comparison with other studies and a detailed discussion of the limitations associated with this research. We find the cost of green hydrogen production of 1.26-1.51 €/$kg_{H_2}$, which is in good agreement with results from ETC of 1.21 €/$kg_{H_2}$[43], Hydrogen council of 1.4 €/$kg_{H_2}$[44] and BloombergNEF of 0.68-1.55 €/$kg_{H_2}$[45]. In contrast to our study, these studies assume an average electricity price and a fixed number of full load hours for electrolysis and thus cannot reflect the system advantage of electrolysis running at very low electricity prices. However, we only assume local learning for electrolysis and do not consider learning due to capacity expansion in other regions, while the other studies analyse global developments. Odenweller et al.[46] show that fast scale-ups of electrolysis capacities until 2030 as we see in our +1.5°C scenarios may be infeasible. However, in this study, we want to show the cost-optimal capacities that are necessary to achieve a given $CO_2$ budget. The demonstrated findings have multiple limitations, firstly in the approach used to model experience curves and secondly in the overall assumptions made for the scenarios. For example, we make the simplified assumption that the electrolysis costs are only influenced by local learning. However, in case electrolysis is scaled up in China or the US, this would also lower the costs in Europe. Furthermore, we assume no imports of hydrogen, which would reduce the need for renewable capacity and electrolysis in Europe.

In this study, we use a sector-coupled European model with endogenous cost reductions through experience curves for electrolysis and renewable generation technologies, assuming different $CO_2$ budgets. In all scenarios, $CO_2$ neutrality is required as an additional condition in the model by 2050. In the second part, we explore the trade-offs of three different methods modelling learning-by-doing.

We find that in scenarios with ambitious climate targets, costs for green hydrogen electrolysis can be reduced to 1.26 €/$kg_{H_2}$ by 2050. In our results, scenarios with a tight budget corresponding to +1.5°C warming require a rapid transformation of the energy system already in 2025 and high build-out rates of both electrolysis and renewable generation. This indicates that, without negative emissions after 2050 or further efforts such as sufficiency measures, the scenarios with +1.5°C warming are difficult to achieve. Hydrogen production shifts from grey to green hydrogen. Depending on climate targets, more than half of hydrogen demand is met by electrolysis in 2025 (+1.5°C scenario) or 2045 (+2.0°C scenario). The option of blue hydrogen is not used in any scenario.

A rapid build-up of both electrolysis and renewable generation capacity is necessary by 2030 to stay within a +1.5−2°C target. While for the former, the European Union's targets for electrolysis are in line with our +1.5−1.7°C budget scenarios, for the latter, the proposed capacities for wind and solar in REPowerEU[6] are behind the capacities necessary to provide sufficient green electricity.

We show that ignoring the virtuous circle between capacity expansion and lower costs leads to delayed investments. A comparison of total costs of scenarios with different transformation speeds should consider cost reductions depending on the usage of technology. A faster expansion of technology than predicted in the exogenous cost assumptions thus leads to lower costs, a slower expansion to higher costs. In our results, total annual system costs are thereby up to 13% higher and levelised cost of hydrogen increase by up to 67% with the exogenous compared to the endogenous method. A middle ground approach where costs are updated sequentially offers the advantage of a lower computational effort compared to the endogenous method and maintains the correlation between investment costs and capacities.

Significant cost reductions in the production of green hydrogen to 1.26 €/$kg_{H_2}$ by 2050 are possible. As further cost reductions and scale-up of electrolysis are expected in the coming years, endogenous cost modelling of electrolysis is important to compare total costs of different scenarios and to determine the right timing for investments.

## Methods
### Model description
We use the open-source European sector coupled model PyPSA-Eur-Sec[30], which minimises total system costs while optimising generation, storage and distribution capacity and dispatch. It covers energy and feedstock demand in the sectors electricity, heating, transport and industry. The model has already been presented in detail in various publications[33,47–49], so in the following only the newly implemented features for this study are presented.

The period from 2020 to 2050 is modelled, with perfect foresight and 7 investment periods at 5-year intervals. Three different $CO_2$ budgets of 25.7, 45 and 73.9 Gt $CO_2$ are assumed for the whole modelling horizon. These budgets correspond respectively to warming of +1.5°C, +1.7°C, +2°C, assuming a per capita share of the global $CO_2$ budgets (further explanations about the chosen budgets are given in an earlier publication[33]). The $CO_2$ emission paths are not fixed, except that $CO_2$ neutrality (or negative emissions) is enforced for 2050, in line with the European Commission's target for net-zero greenhouse gas emissions.

Existing generation capacities (lignite, coal, gas, hydro, nuclear, solar, offshore and onshore wind) are taken from the IRENA 2020 report[9] and the open-source package powerplantmatching[50]. Costs, lifetime and efficiencies are assumed for the respective year of the Danish Energy Agency DEA[31]. The investment costs of renewable generation capacity (solar, onshore and offshore wind) and hydrogen electrolysis are in the endogenous case not exogenously specified but are part of the optimisation in form of a piecewise-realised one-factor

experience curve. This results in a mixed integer linear problem MILP which is further described below. The total system costs are not discounted by a social discount rate that reflects the value of future investments in order to investigate the impact of learning in isolation from other effects. A high discount rate leads to a significantly lower weighting of costs in 2050 and thus shifts investments into the future while endogenous learning leads to earlier investments.

There are three different competing options for producing hydrogen in the model: (i) grey hydrogen (via steam methane reforming (SMR)), (ii) blue hydrogen (SMR + carbon capture) and (iii) green hydrogen (via electrolysis). Hydrogen can be used for methanation, for heating (hydrogen boilers), electricity (fuel cells and retrofitted OCGT), in the industry, and in the transport sector. The demand pathway for hydrogen in parts of industry and transport is exogenously defined, while in all others sectors, hydrogen competes with other ways of supplying demand in these sectors. This means both demand for hydrogen (e.g. where it competes with heat pumps in the heating sector) and the supply side, for example if the hydrogen is produced via electrolysis or SMR, are part of the optimisation. OCGT and gas boilers for heating can be retrofitted to run flexibly with natural gas or hydrogen (Fig. 1). There are various electrolysis technologies. In this study, cost and efficiency assumptions of alkaline electrolysis cells (AEC) are used since they are currently the most common electrolysers available on the market.

In contrast to other studies, no maximum annual expansion rate of renewable generation capacity and no annual $CO_2$ emission paths are specified. Those constraints often reduce strongly the solution time of the optimisation problem, but also predetermine the transition paths. Since we consider multiple sectors and investment periods, the optimisation problem is aggregated spatially and temporally to make it computationally solvable. Spatially, energy transmission networks are reduced to a single node for Europe, while six different typical regions are used to represent the variability of renewable generation. We optimise the power transfer capacity between transmission and distribution level. No existing grid infrastructure of distribution grids is assumed. Losses in distribution are neglected. Costs of the distribution grid of 500 €/kW$_{elec}$ are applied. Electricity demands, heat pumps, resistive heaters, rooftop PV, home batteries, and electric vehicles are connected to the distribution grid. All remaining technologies of the power sector (e.g. large scale storage, wind parks, conventional power plants, electrolysers) are connected to the high-voltage grid. In the Supplementary Material, we analyse the impact of spatial aggregation on our results by comparing scenarios with higher spatial resolution and the exogenous method with our results from the manuscript (see Fig. S17a). The infrastructure cost contributes 0.1–7.6% to the total annualised costs only by a small margin. In terms of temporal aggregation, 10 typical days per investment period are considered. This allows, in contrast to most IAMs, the possibility to

represent the temporal variation of the renewable generation and is comparable to other ESM like PRIMES using two or nine typical days[51] or Heuberger et al.[28] using 11 temporal days. The 10 typical days are obtained through k-medoids clustering using the Python package tsam[52,53], so that they represent the average statistics of weather and demand, while also capturing more extreme events. To find the optimal solution we use the commercial solver Gurobi[42] using 12 threads.

### Endogenous learning
Experience curves are an economic concept based on empirical evidence in which the specific investment costs $c$ decrease by a constant factor $\alpha$ with each doubling of experience $E$

$$c(E) = \overline{c_0} \cdot \left( \frac{E}{\overline{E_0}} \right)^{-\alpha} \text{ with } \alpha \text{ given by } \alpha = \log_2 \left( \frac{1}{1 - \text{LR}} \right). \qquad (1)$$

The constants $\overline{c_0}$ and $\overline{E_0}$ are fixed starting points, LR is the so-called learning rate (Table 2). If for example the learning rate is 20% (LR = 0.2), the costs are reduced by 20% for each doubling of cumulative experience. Typically learning rates range between 5%-25%. Smaller modular technologies (e.g. PV or wind) tend to have higher learning rates than large-scale plants[54,55]. In this study, the global cumulative capacity is used as a proxy for experience. $c$ represents the investment costs [EUR per MW], $c_0$ the initial investment costs [EUR per MW] Tables 1 and 2.

Experience curves make ESM optimisation problems both non-linear and non-convex, which makes solving particularly challenging. There are two main approaches[25] to integrate experience curves within ESM: direct non-linear implementation and piecewise linear approximation of cumulative costs. In this paper, we follow the latter and use special ordered sets of type 2 (SOS2). Compared to the non-linear implementation this has several advantages: it can find a global minimum rather than getting stuck in a local one, it does not depend on the initial starting point of the solver, and it can be solved faster using commercial solver algorithms.

**Total technology cost TC**. The cumulative investment costs for technology can be obtained by integrating the experience curve (Equation 1). For $\alpha \neq 1 \rightarrow \text{LR} \neq 0.5$ this results in

$$\text{TC} = \int_{E_0}^{E} c \, dE' = \frac{1}{1 - \alpha} \left( c(E)E - \overline{c_0 E_0} \right). \qquad (2)$$

The cumulative costs TC are stepwise linearised with a given number of line segments (Fig. 7). The greater the number of segments, the more precise the solution. However, this also increases the number of variables and thus the solution time.

**Table 2 | Summary of parameters (left) and variables (right)**

| Parameter | Definition | | Variable | Definition |
|---|---|---|---|---|
| $\overline{c_0}$ | initial investment costs | | c | investment costs |
| $\overline{E_0}$ | initial experience | | E | experience, cumulative installed capacity |
| $\alpha$ | learning index | | TC | cumulative technology costs |
| LR | learning rate | | $\delta$ | continuous variables $\in [0, 1]$, part of SOS2 |
| s | technology (e.g. solar) | | cap | new build capacity per investment period |
| t | investment period | | inv | investment cost per investment period |
| i | interpolation point | | | |
| N | total number of interpolation points used for piece-wise linearisation | | | |
| gf | global factor, share of global capacity installed in Europe | | | |
| $(\overline{E_i}, \overline{TC_i})$ | interpolation points of piece-wise linearisation | | | |
| $m_{s,j}$ | slope of line segment $j$ for technology $s$ | | | |

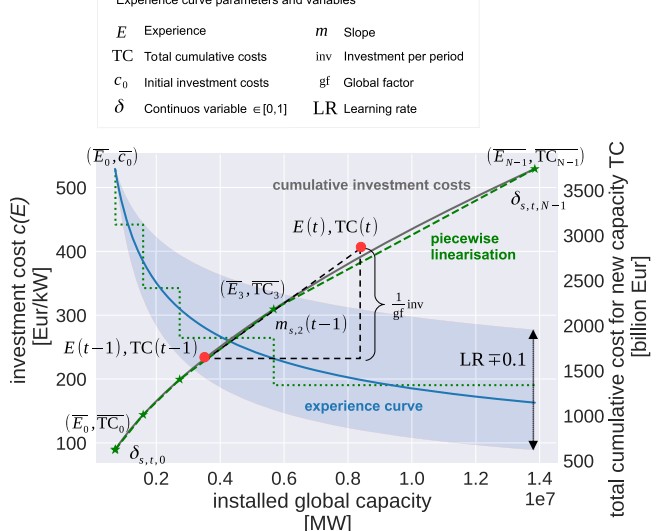

**Fig. 7 | Experience curve (blue) and cumulative cost curve (grey), as well as the implemented piecewise linearisation (hashed green line) with in this case four line segments.** The shaded blue area shows the variation of the learning rate LR by ±10%. The experience $E$ is defined as the cumulative installed capacity. Investment costs $c$ are decreasing with increasing experience. From the slope $m$ between the experience $E$ and the total cumulative cost TC at two different time steps, we calculate the investment per period inv. Since we model only Europe, global developments are represented by the global factor gf.

Other models use the following number of segments: In ERIS[56] 6, global MARKAL 6–8, MARKAL-Europe 6–20[57]. Heuberger et al.[28] 5. In this study we assume five line segments. This corresponds to six interpolation points $(\overline{TC_i}, \overline{E_i})$ (marked in Fig. 7 with a star) for $i \in [0, N\text{-}1]$ where $N$ is the number of interpolation points.

The definition of the line segments follows Barreto's approach[27]: line segments at the beginning of the learning are shorter to capture the steep part of the experience curve more precisely. With each line segment the cumulative cost increase doubles.

**Special ordered set of type 2 SOS2.** We define a set of continuous variables $\delta_{s,t,i} \in [0,1]$ for each technology $s$ (e.g. solar PV), investment period $t$ and interpolation point $i \in [0, N-1]$ of the linearisation. $N$ is the total number of interpolation points used for the piece-wise linearisation. The continuous variable indicates which line segment is active. Only two adjacent delta variables are non-zero at the same time $t$ for technology $s$ and their sum is equal one

$$\sum_{i=0}^{N-1} \delta_{s,t,i} = 1. \qquad (3)$$

For example, if line segment $i$ is active then only $\delta_{s,t,i}$ and $\delta_{s,t,i+1}$ are non-zero.

**Cumulative experience $E$.** The cumulative capacity $E_{s,t}$ of a technology $s$, time $t$ and interpolation points $i$ is defined as a summation of the product of the continuous variable $\delta$ and the x-position of the interpolation points $\overline{E_{s,i}}$

$$E_{s,t} = \sum_{i=0}^{N-1} \delta_{s,t,i} \cdot \overline{E_{s,i}}. \qquad (4)$$

For example, if line segment $i$ is active, then (4) will interpolate between $\overline{E_{s,i}}$ and $\overline{E_{s,i+1}}$.

The new installed capacity $\text{cap}_{s,t}$ per investment period in Europe is the difference of the cumulative experience weighted by a global

factor gf

$$\text{cap}_{s,t} = \text{gf} \cdot (E_{s,t} - E_{s,t-1}). \qquad (5)$$

The global factor is one for local learning (assumed for hydrogen electrolysis in this study). For global learning (as considered for PV, onshore and offshore wind in the following results) it represents today's share of European compared to global capacities.

**Linear combination.** If no time-delay for the learning effects would be considered, one could express the cumulative cost TC similar to the cumulative experience $E$ with the help of the SOS2 variables $\delta$ and the y-position of the interpolation points $\overline{TC_{s,i}}$ as

$$TC_{s,t} = \sum_{i=0}^{N-1} \delta_{s,t,i} \cdot \overline{TC_{s,i}}. \qquad (6)$$

**Temporal-delayed learning effect.** In this study we consider a temporal-delayed learning effect which means that the investment cost decrease in an investment period $t$ depends on the cumulative installed capacities at the previous investment period $t-1$. This represents learning effects more realistically, as investment costs do not decrease immediately in the same reference year as the emerging technology is employed, but are subject to a time lag. One should be aware that this results in an overestimating of the cumulative cost curve shown exemplary in Fig. 7. The cumulative costs are defined with the delayed learning as

$$TC_{s,t} = TC_{s,t-1} + m_{s,t-1,i} \cdot \text{cap}_{s,t}. \qquad (7)$$

Here $m_{s,t-1,i}$ is the slope of the line segment at the previous investment period $t-1$. As the total costs are the integral of the experience curve, the slope $m_{s,t,i}$ is equivalent to the specific investment costs.

The overall investment costs per investment period $\text{inv}_{s,t}$ are defined as

$$\text{inv}_{s,t} = \text{gf} \cdot (TC_{s,t} - TC_{s,t-1}). \qquad (8)$$

## Data availability
The raw and processed data used in this study are archived at Zenodo https://zenodo.org/record/6645232#.Yt6L6tJBwkI under a CC-BY-4.0 license.

## Code availability
The PyPSA-Eur-Sec model is available under MIT license via Github https://github.com/PyPSA/pypsa-eur-sec. Model documentation https://pypsa-eur-sec.readthedocs.io/en/latest/. All the technology assumptions are available via Github https://github.com/PyPSA/technology-data, version v0.3.0 is used in this study. The source code and input data for this study are openly available at Zenodo https://zenodo.org/record/6645232#.Yt6L6tJBwkI under a CC-BY-4.0 license.

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

## Acknowledgements
The authors thank Johannes Hampp, Falko Ueckerdt, Markus Millinger, Lina Reichenberg, Fredrik Hedenus and Niclas Mattsson for helpful discussions and suggestions. T.B. and E.Z. acknowledge funding from the Helmholtz Association under grant no. VH-NG-1352.

## Author contributions
E.Z.: Methodology, Software, Validation, Formal analysis, Investigation, Data Curation, Writing - Original Draft, Visualisation, Conceptualisation MV: Writing- Review & Editing, Supervision, Conceptualisation, Methodology T.B.: Conceptualisation, Methodology, Resources, Writing- Review & Editing, Supervision, Funding acquisition, Project administration.

## Funding

## Competing interests
The authors declare no competing interests.
