## [Peer Review File · Nature Communications]

REVIEWER COMMENTS

Reviewer #1 (Remarks to the Author):

The present study shows the impact of endogenously applying technological learning for alkaline electrolysis to European energy system modelling. Therefore, it points out potential shortcomings of projections of future renewable hydrogen integration in system models that rely on exogenously defined cost developments and/or market potentials. However, in my opinion, the study misses some important aspects and needs some revision before being accepted for publication. Detailed remarks are found in the following.

General:

- Structure:

The article misses a general structure (e.g., Introduction/Results/Discussion/Methods, as found in the author guidelines) and so it is partially hard to follow or distinguish actual outcomes and results from the rest. However, it seems to basically follow the proposed structure and maybe just misses some major headings.

- "learning curve" vs "experience curve":

Even though both terms are often used interchangeably in relevant literature, they originally have different meanings: the learning curve applies to a single production process or company, whereas the term experience curve extends the scope of the effect to an entire industry (<https://doi.org/10.1016/B978-0-12-818762-3.00002-9>). So I would suggest staying with the term "experience curve" in your article (and/or point out the differences).

Section "Main":

I would suggest swapping the first two paragraphs for a more continuous reading (1. hydrogen demand and its implications in Europe - 2. introduction on hydrogen from water electrolysis and potential scaling effects - 3. how this is discussed in literature so far).

p. 1, 3rd paragraph: "Several studies have been carried out on how hydrogen electrolysis investment costs develop over time or with capacity deployment" - The discussed studies only cover projections on future costs. If this was the intention, it should be made clear in the mentioned statement, otherwise also analysis of past cost developments should be included and discussed (e.g., <https://doi.org/10.1016/j.ijhydene.2008.03.011>, <https://doi.org/10.1038/nenergy.2017.110>, <https://doi.org/10.1016/B978-0-12-818762-3.00010-8>). Even more so, as they represent the basis for electrolysis learning rates.

p. 2, 2nd para: Regarding endogenous learning in energy models, there is at least some research for the transport sector as well (e.g., <https://doi.org/10.1016/B978-0-12-818762-3.00015-7>).

p. 3, 1st para: I think it is fair in the scope of this study to not distinguish between different electrolysis technologies, especially since all types will be affected to some point due to spillover learning effects. However, as shown in <https://doi.org/10.1016/j.ijhydene.2019.09.230> and <https://doi.org/10.1016/j.apenergy.2020.114780>, there are significant differences in how the individual technologies are affected by scaling (learning and upscaling). Thus, this should at least be discussed.

You include "Methane" production as a potential usage of hydrogen. However, it is not indicated, how this methane (SNG) is used and if it could potentially be substituted by hydrogen. As methane production is found to be the main path in the 1.5°C scenario this needs to be discussed properly.

Section "Electrolysis investment costs":

Even though the reference you used for setting the initial electrolysis costs is basically comprehensible, they seem rather low compared with other literature (e.g., <https://doi.org/10.1016/j.apenergy.2020.114780>, https://www.irena.org/-/media/Files/IRENA/Agency/Publication/2020/Dec/IRENA_Green_hydrogen_cost_2020.pdf, <https://www.ipa.fraunhofer.de/content/dam/ipa/de/documents/Publikationen/Studien/Studie-IndWEDe.pdf>). While it becomes more reasonable knowing that these values are for capacities of 100MW per system, this should at least be stated. As to my knowledge, there is no system of this size in operation up to now, so it is rather a hypothetical value. Furthermore, as you only focus on alkaline electrolysis (which is also the cheapest so far) as the "general type" in your analysis, taking this rather low (related to all types) starting value, adds some significant bias. Even more so, as this impacts the overall electrolysis cost level in the whole model. Therefore, a proper sensitivity analysis on the impact of the initial system costs is a must in my opinion.

Section "Comparing endogenous and exogenous learning":

p. 9, 1st para: You should probably recap that technological learning occurs as a function of produced (cumulative) volumes and not as a function of time. Thus, the sequential method is more accurate than the completely exogenous model (which would still presume cost reductions even without any production).

p. 11, top: "... the total annualised costs are overestimated by 9% and investments in emerging technologies are delayed." - this is probably dependent on how the threshold for the iterations is set.

Section "Limitations of this study":

- the model only considers production-related scaling effects, effects of unit scaling (individual scales of the installed electrolysers) are neglected
- spill-over effects between electrolysis (see above)

"In all scenarios, the electricity generation from renewables is greater than the electricity demand of the electrolysis at every point in time. It is therefore possible, that all hydrogen produced via electrolysis is green..." - I assume that "at every point in time" narrows down to every calculated year. So this statement is true for "on balance". There may be times, where storage is needed for a continuous production or conventional generation, due to non-correlation supply and demand.

Reviewer #2 (Remarks to the Author):

● Summary

- This paper explores the growth potential of hydrogen technologies to support decarbonization.
- The focus of the paper is on the impact of implementing learning-by-doing versus exogenous cost estimates for planning modeling.
- Findings show that including learning-by-doing in the planning process accelerates the uptake of hydrogen technologies, lower cost of hydrogen and lower overall system cost.

I have identified several issues and included questions to address those issues below.

- **Page 2:** The authors mention that a comparison of 15 IAM found that costs have fallen below 2050 expectations. There is no reference included. Does that mean it was done as part of this work? I follow that those findings highlight the need to avoid setting high floor

costs, but that statement alone does not show the need to remove restrictions on expansion rates or max feed-ins. Maybe there is another reason for limiting expansion rates in modeling (subsidy limitations, manufacturing limitations, or building queue limitations). This sentence should be reworded so as not to overstate the findings from the IAM comparison. If the authors want to make the points regarding expansion rates and max feed-ins please include additional references to support that statement.

- Page 2: "This highlights the need to model investment costs endogenously without assuming too high floor costs, restrictions on expansion rates or maximum feed-ins of renewable energies."
 - Introducing constraints on expansion rates or max feed-in do not necessarily change the results. I agree that removing those constraints can improve the model; however, you make it sound like this is a critical problem that needs to be solved. Particularly when we are looking at a trade-off between computational time and additional features or resolution, why do you think this is important? I can see a scenario where the expansion rate and max feed-in are appropriately set so they don't affect the outcome of the optimization and instead the computational capacity is used to increase spatial resolution resulting in potentially more accurate results. Please support why you think this is important to remove these constraints versus improving other features.
 - Word choice for "too". Change to "excessive" or something similar
- Page 2: "We do not specify the expansion rates of renewable capacities nor CO2 emission targets for individual years. These additional conditions often lead to a lower computational effort but also predetermine the transition paths."
 - The use of the phrase expansion rate without additional qualification is confusing. In this case I assume you are referring to maximum annual rate but please update to make this clear.
 - Secondly, if you are referring to max annual expansion rate, you have not made the case for why this is important. There are practical factors that can limit max annual installation capacities (building permits, subsidies/credits, installation time, etc.). You have not sufficiently justified that free expansion is appropriate.
 - As mentioned before, from a modeling perspective if you set the max expansion rate sufficiently high it will not affect the results as the authors have suggested; however, this exception was not mentioned.
 - Lastly, despite the discussion in the paper about the importance of relaxing expansion rates, you performed a sensitivity in which build out rates of renewable generation capacity are limited. I am seeing mixed signals in your narrative regarding the most appropriate way to proceed and there is no justification either way. Please justify the relaxation of max annual capacity limits.
- Page 3: "cost and efficiency assumptions of alkaline electrolysis cells (AEC)"
 - While alkaline is the market leader now, PEM is rapidly growing. It is not readily apparent that alkaline will be the market leader in 2050.
 - Including detailed learning algorithms is a good way to explore the potential for other competing electrolysis technologies. The authors need to justify the use of only alkaline by discussing PEM costs expectations and discussing how their inclusion might impact the results.
- Page 3: figure 1: Use different line styles (pattern, color and/or thicknesses) for the fully exogenous and mixed lines. It is unnecessarily difficult to distinguish between them.
- Page 4: figure 2: Please change the legend to better define the scenarios in figure 2. 1p5 should be +1.5°C, or something similar. 1p5 is never defined -- we are left to infer.
- Page 4: figure 2: You mentioned that SCC can drive the total cost for the +1.7C scenario below the +2.0C scenario, but figure 2 caption specifically says (without climate damage included) so please describe why the +1.7C goes below the 2.0C scenario in 2050 in figure 2. This directly reflects on the credibility of the model and should be discussed.
- Page 6: What is included in the investment cost? (stack, balance-of-plant, installation, land, grid connection) Additionally, you need to mention how operating and maintenance

costs are included in the main text. I see from the supplemental that a value is provided for electrolyzer but what is included in that value (labor, water, stack replacement, etc.)?

- Page 6: The supplemental material includes costs for electricity distribution grid and grid connection, but it is never mentioned how those are used. Are these flat costs added to the investment cost? Do those electricity infrastructure items benefit from learning as well?

- Page 6: Readers can benefit from seeing the typical breakdown of hydrogen production by end use. This is not well described in the text but it is presented in the supplemental. Please either add a figure showing a representative example breakdown of end uses for hydrogen or add additional text describing the general behavior and a reference to the supplemental material for more information.

- Page 6: The finding that no blue hydrogen is used is an important finding but only receives a few sentences and an additional sensitivity scenario. The additional scenario performed is one in which SMR+CC benefits from endogenous learning while electrolysis and renewables does not. This is not enough to increase use of blue hydrogen, but that is a really weak scenario. The dynamics of what is happening are not explained. I am left to assume that the cost for SMR+CC is greater than electrolysis+renewables even in the sensitivity, but it is never stated by the author. Please add discussion about the cost dynamics that result in this result. Also, if you never see blue hydrogen, how are you sure it is working correctly. As a reader, I'm not sure that your model correctly includes blue hydrogen. The sensitivity analysis should have instead looked at the price point at which we start seeing blue hydrogen to provide some context for why it isn't included. Only performing a scenario with endogenous learning added to SMR+CC does not adequately address the subject of blue versus green hydrogen.

- Page 7: Similar to the question about electrolyzer costs, what is included in the renewable cost? Is this overnight costs? What is included in O&M?

- Page 7: There is some effort to compare the results to previous work (e.g., REPowerEU plan), but this is only done for specific results and not across all of the scenarios (e.g., not done for the +1.5C case). As such, I'm not sure how realistic and reasonable these results are. That in turn, limits the usability and impact of the results. Without a more detailed comparison, this is a nice technical study but not actionable from a system planning or electrolyzer manufacturer point of view. Please include more comparisons to related work to understand the results in the context of other models. I understand that no other study has introduced learning-by-doing with hydrogen, but there are, in addition to the one that you provided, other studies that look at achieving climate targets or high renewable share scenarios (without hydrogen or a limited representation of hydrogen) that could be used to provide some insight into the magnitude of the renewable capacity needed which could be compared to your results.

- Page 8: The "seqcost" scenario is introduced but not described in that section or the supplemental. I finally found out that it was sequential cost method on page 9, but the term "seqcost" is not included in the paper text at all. Please add a description of the "seqcost" scenario near its first use on page 8 and also in the supplemental. Not being able to search for it in the text cost me 10 minutes to figure out what "seqcost" meant.

- Page 9: Please add a description for why "Seqcost" is important. As I understand it, the seqcost scenario essentially allows learning to be applied to the technologies while maintaining linearity. Does that accurately capture the meaning of seqcost? If so, why are the results not more similar to the endogenous scenario? I think the reader could benefit from more information on how the investment cost is changed (because it is clearly a function of capacity so it must be based on the learning rate since endogenous is only based on time and not capacity). I assume that the seqcost utilizes the same learning rate expression and value that the endogenous scenario does, but the author needs to include more details about how the seqcost scenario is implemented? Finally, what threshold was used and is it an important assumption?

- Page 9: Hydrogen costs represent production and storage costs, right? I have not seen any discussion on the distribution/delivery costs. Please describe how those costs are

considered in your work. There is one region so am I left to assume that there is no cost for hydrogen distribution and delivery? Further, hydrogen storage costs are included in the supplemental material of table 2 but it is never discussed how the hydrogen storage is sized. Please describe how hydrogen storage is sized.

- Page 9: The "Comparing endogenous and exogenous learning" should be reworked to focus more on a comparison of the three methods. At present it provides some anecdotal explanation of the results based on the selected exogenous curve. Assuming a low learning rate would change the magnitude of electrolysis installed, the cost of hydrogen and the total system cost, so the authors should speak more generally about the endogenous versus exogenous results than on the baseline learning rate results. Overall, I think the paper is missing discussion about the merits of the different methods (endo., exog., and seqcost) and that is what I would like to see included once this section is reworked.

- Page 10: why aren't there bars for seqcost in 2020 and 2025? Is there no difference or were those left out? Please print the values for each bar on the chart to make it clear.

- Page 10: How can figure 6 be an absolute value, while the other figures include plus or minus 10%. I am left to assume that the difference uses the base endogenous scenario. Please add the results for plus and minus 10% learning rate for the endogenous scenarios to the figure. Also the authors have identified that the +1.5C is difficult to reach so why is it the only comparison point between methods. I need to see the results for +1.7C and +2.0C to be able to discern the importance of implementing an endogenous learning strategy.

- Page 10: Similarly, I think figure 6 is one of the main outcomes of this work but it is not well described or put in context with the tradeoffs between cost, technology delay, computational benefit and spatial resolution. Only one instance is shown for endogenous versus seqcost (21hrs versus 1hr, with a 9% overestimation when compared to the base endogenous scenario with +1.5C.

- First, based on the values in figure 6 what does the 9% value in the text represent? No values are greater than 7% in figure 6.

- Second the authors need to include a more comprehensive discussion about the tradeoffs between methods. It appears that in 2050 the energy balances between the three methods are quite similar (fig 10-13). For the +1.5C case, if the energy balances are quite similar in 2050 (fig 13e) and the

- You can't relate them since the costs per unit are not comparable

- Page 11: How are FOM costs affected by learning? More systems operating is likely to materially affect maintenance costs. I assume like efficiency and lifetime, learning is not applied to FOM. If that is right, please include in limitations.

- Page 11: What can you say about the relative impact of the limitations? What should be the highest priority targets items to include in the next version of the model (e.g., 1) endogenous efficiency, 2) endogenous lifetime, 3) greater spatial resolution, ...)?

- Page 11: Regarding spatial resolution, I am particularly interested in the impact of assuming that there is only one region in Europe. Please provide a sense for the impacts on system cost for this assumption. I think this is important because lots of effort is put into including endogenous cost estimates, but if the spatial simplification greatly underestimates costs, then it might be better to introduce seqcost method with multiple regions in Europe. I recognize that this may be beyond the scope of the analysis; however, when the authors make such strong statements about why it is important to include endogenous cost estimates, then it is necessary to support their claims by considering the tradeoffs between alternative strategies. (also see spatial resolution comment for page 14)

- Page 12: At several points in the paper (including page 12), the author shows that exogenous cost estimation methods lead to higher costs, but this result is somewhat anecdotal and a function of the learning rate selected compared to the exogenous cost estimation. A low learning rate could be worse than the exogenous case, and an optimistic exogenous cost curve could be lower system cost than the baseline endogenous estimate. To that end, the discussion should focus more on the strengths and weaknesses of each of the methods and not focus on their resulting values. Implementing an endogenous method

for assessing costs is not inherently better, but it represents a tradeoff between different aspects. This will require that you rework the "Comparing endogenous and exogenous learning" section the "limitations" section and the "conclusion" section to focus more on tradeoffs.

- For example, the fourth paragraph of the conclusion discusses that "ignoring the virtuous cycle... leads to severe distortion of the results"; however, this paper only shows one example of this based on a range of learning rates and one set of exogenous prices from DEA and does not provide sufficient verification of these results.
- The last paragraph of the conclusion is not something that you have proven from your work. The first sentence is speculation and not specific enough to be verified. The other two sentences are not adequately proven by this paper. There was a lot of modeling work that was done, but the scenario development, sensitivities and discussion in this paper do not work together to verify the hypothesis in the last paragraph of the conclusions. I personally understand the value of integrating endogenous cost estimates into these models but you have not made a clear and strong case for that.
- Page 14: "...no maximum build out rate of renewable generation capacity and no annual CO2 emission paths are specified but also predetermine the transition paths"
- See similar comment from text on page 2. Please reword to clarify.
- Page 14: "To make the optimisation problem still computationally solvable, it is aggregated spatially and temporally" a single node in Europe
- It sounds like you are trading a max buildout rate constraint for severely limited spatial resolution. This renders energy transmission infrastructure immaterial. Distribution and transmission infrastructure is an important feature for the cost and reliability of electricity, gas, and hydrogen networks. I see that you include some variability for renewables (6 profiles), but a key unanswered question is how the transmission networks evolve to meet future production and demand of electricity, natural gas and hydrogen.
- Please justify your simplification of the model to one node for Europe.
- Supplemental
- Several figures in the supplemental material are too small to read. Please increase the figure size and text size to make all figures legible.
- Figures 10-13 need unique and detailed captions.
- Figure 11 f is incorrectly repeated from fig 11 d
- The table starting on page 16 does not have a caption. Also, there are no units on that table so I can't tell if FOM values are absolute or percentage of capital cost. Please add units for every row in the table.
- I mentioned in an earlier comment but it is not clear what is included in the "investment" cost. Since you didn't say capital cost, I assume it is capital and installation at least. Please add a paragraphs of text before the table describing some of the details of what is included in the investment cost.
- Please add references for where each cost item comes from in the second table.
- Specifically for electrolysis costs, FOM is 2 (€/kW-year, I presume), which when compared with the investment cost of €650/kW in 2020, over a 25 year timeperiod it appears that no stack replacements will be performed or at least they aren't part of the FOM. Readers need the reference to be able to understand what is included. Because of the importance to the results, please justify the values selected for electrolysis and compare with other sources.
- In the second table not all technologies have a lifetime. Why not, and if no lifetime is included, then what is assumed?
- Because of all of the legend entries, Figures 10-13 are not particularly useful. You can keep the figures in the supplemental, but I recommend inserting the values in table form as well. Readers will be able to explore this data more easily than trying to compare across figures. Also, you should include installed power in addition to a table that shows energy.
- The supplemental has a good discussion section for each figure until the "energy balance" section, after which, there is no discussion for any of the subsequent figures. Given the

importance of figures 10, 13, 15-16, the reader would benefit from some discussion for those figures.

- **Summary**

- **This paper provides a nice but relatively narrow improvement to the implementation of cost estimation in modeling.**
- **There is a lot of discussion on the absolute results from the runs, but given the limitations in the assumptions it will be difficult for system planners, technology investment groups or technology manufacturers to take actionable steps from this work. That said, I think the paper should focus on the tradeoffs between different cost estimation methods by providing a more comprehensive discussion.**
- **I recommend major revisions and would consider accepting the paper if my questions and comments are addressed adequately.**

Reviewer #3 (Remarks to the Author):

The paper for review "Endogenous learning for green hydrogen in a sector-coupled energy model for Europe" presents a PyPSA model analysis and results of the European energy system, filling knowledge gaps in literature, noting other studies have either left out important sectors of hydrogen demand, ignored the temporal variability in the system or neglect the dynamics of learning effects.

It is highly likely that costs and efficiencies will improve as production scales up and this study captures some of the dynamics of this for hydrogen, noting if electrolysis costs are modelled without dynamic learning-by-doing, then the electrolysis scaleup is significantly delayed, while total system costs are overestimated.

The study indicates that without negative emissions after 2050 or further efforts such as sufficiency measures, the scenarios with +1.5oC warming are difficult to achieve, which is disappointing. Is this attributed to just the build out rates of solar wind and electrolysis? I would suggest to find more references agreeing with or opposing the points made by Way et al. [8], so it would support your result or not.

It seems that 1.7degreesC is achievable more so than 1.5degreesC and less emissions than 2degreesC, could this be a useful recommendation from this study?

The limitations noted are fair and welcome that they are noted.

The study is laid out well, explained to an appropriate level and presented in a clear concise way. The methodology is sound from the information provided.

We suggest the authors to check the numbers in this section: "between 171-237 e/kW for solar PV, 818-900 e/kW for onshore wind and 1327 e/kW - grid connection costs are added to these investment costs in the model that do not undergo any learning"

We suggest to edit the text as there is a double link added in ref 27.

Regarding supplementary data Table 6 on page 16, 17 & 18 is not labelled and you only have reference 5 as the sole source of data; which is from 2019. Is this prudent? Should multiple sources be used to align expected costs and technology?

Also add a number of acronyms to the list e.g. FOM & VOM

We thank the authors for their interesting work.

RC: Reviewer Comment, AR: Author Response, █ Manuscript text

We thank the reviewers and editor for considering our research article and giving very detailed and very useful feedback, that has allowed us to improve the manuscript. Please find below the point by point responses to the reviewer comments. The differences compared with the previous submitted version of the paper are highlighted in blue and red text in an attached file.

1. Reviewer #1

RC: The article misses a general structure (e.g., Introduction/Results/Discussion/Methods, as found in the author guidelines) and so it is partially hard to follow or distinguish actual outcomes and results from the rest. However, it seems to basically follow the proposed structure and maybe just misses some major headings.

AR: We added major headings in the revised version to clearly distinguish the introduction, results, limitations of the study and the conclusion.

RC: Even though both terms are often used interchangeably in relevant literature, they originally have different meanings: the learning curve applies to a single production process or company, whereas the term experience curve extends the scope of the effect to an entire industry (<https://doi.org/10.1016/B978-0-12-818762-3.00002-9>). So I would suggest staying with the term "experience curve" in your article (and/or point out the differences).

AR: In the revised version we only use the term 'experience curve' instead of 'learning curve' in order to follow the original correct definition of the two terms.

RC: I would suggest swapping the first two paragraphs for a more continuous reading (1. hydrogen demand and its implications in Europe - 2. introduction on hydrogen from water electrolysis and potential scaling effects - 3. how this is discussed in literature so far).

AR: We have restructured the introduction. As suggested by the reviewer, in the revised version we swapped the paragraphs and start with today's hydrogen demand in Europe and possible hydrogen production processes, followed by an introduction to green hydrogen and potential learning effects.

RC: p. 1, 3rd paragraph: "Several studies have been carried out on how hydrogen electrolysis investment costs develop over time or with capacity deployment" - The discussed studies only cover projections on future costs. If this was the intention, it should be made clear in the mentioned statement, otherwise also analysis of past cost developments should be included and discussed (e.g., <https://doi.org/10.1016/j.ijhydene.2008.03.011>, <https://doi.org/10.1038/nenergy.2017.110>, <https://doi.org/10.1016/j.ijhydene.2017.110>).

[//doi.org/10.1016/B978-0-12-818762-3.00010-8](https://doi.org/10.1016/B978-0-12-818762-3.00010-8)). Even more so, as they represent the basis for electrolysis learning rates.

AR: We have referred here to publications with future cost developments of electrolysis, as we want to point out the problem of predefined production or capacity developments or expert surveys in these. The assumptions about future developments can significantly influence the results. This is not a problem for publications on historical cost development to e.g. determine the learning rate. In the revised version, we have clarified that these are studies that deal with future and not historical cost developments.

RC: p. 2, 2nd para: Regarding endogenous learning in energy models, there is at least some research for the transport sector as well (e.g., <https://doi.org/10.1016/B978-0-12-818762-3.00015-7>).

AR: The proposed article (<https://doi.org/10.1016/B978-0-12-818762-3.00015-7>) for endogenous learning in the transport sector refers to an Integrated Assessment Model (IAM). Our statement that previous studies either consider only a single country or only the electricity sector refers to Energy System Models (ESM) as described at the beginning of the previous sentence. IAMs consider endogenous learning for different sectors. We have clarified this in the introduction.

p. 2

■ *Endogenous cost reductions are currently used in some IAMs [18-21] which consider learning in multiple sectors and global developments.*

RC: p. 3, 1st para: I think it is fair in the scope of this study to not distinguish between different electrolysis technologies, especially since all types will be affected to some point due to spillover learning effects. However, as shown in <https://doi.org/10.1016/j.ijhydene.2019.09.230> and <https://doi.org/10.1016/j.apenergy.2020.114780>, there are significant differences in how the individual technologies are affected by scaling (learning and upscaling). Thus, this should at least be discussed.

AR: We assume in this study alkaline electrolysis cells (AEC). We only consider large plants (above 100 MW) to avoid the scaling effects of very small plants. We have added a paragraph in the Supplementary Material discussing different electrolysis technologies and refer to this analysis in the main manuscript. In this section, we discuss the learning rates of the three electrolysis technologies AEC, PEM and SOEC, as well as their differences in terms of unit scaling. In a sensitivity analysis, we vary the investment costs for electrolysis in 2020 across the range of cost assumptions of the three different types of electrolysis (650-5850 EUR/kW_{elec}). By varying the investment costs, hydrogen continues to be produced by electrolysis and the investment costs reduce over the modelling horizon. With higher initial investment costs in 2020 of 1300 EUR/kW_{elec}, which is in the range of PEM investment costs in 2020, investment cost decrease to 189 EUR/kW_{elec} in 2050. The hydrogen volume and usage is comparable to our base assumptions for investment costs in 2020 in the range of 650-1300 EUR/kW_{elec}. With higher initial costs the production of grey hydrogen increases and the usage of hydrogen is reduced by up to 15% in 2050 with initial cost assumptions of 5850 EUR/kW_{elec}. We are mentioning in the main manuscript the limitation that we do not consider spill-over effects between electrolysis technologies.

p.15

■ *No spill-over effects between technologies are taken into account. These effects could be important for example for on- and offshore wind, but also for spill-over between different electrolysis types.*

RC: You include "Methane" production as a potential usage of hydrogen. However, it is not indicated, how this methane (SNG) is used and if it could potentially be substituted by hydrogen. As methane

production is found to be the main path in the 1.5°C scenario this needs to be discussed properly.

AR: Thank you very much for the valuable advice. We have both added (i) in the introduction the possible uses of synthetic gas and (ii) in the result section in which areas synthetic gas is used and in what proportion.

p.3

■ *The synthetic gas from the methanation can be used in the heating sector (gas boilers or combined heat and power plants (CHPs)), for industry processes or in the power sector (OCGTs or closed cyclic gas turbines CCGTs).*

p.8

■ *In the +1.5°C scenario, the hydrogen is primarily used to produce synthetic fuels (1800 TWh_{H₂} in 2050) and synthetic methane (700 TWh_{H₂}/ 560 TWh_{CH₄} in 2050). The produced synthetic methane is mainly used in industry process (424 TWh_{CH₄}) but also in OCGTs (135 TWh_{CH₄}). There is an option to convert the OCGTs to run on hydrogen, but this option is not used.*

RC: Section "Electrolysis investment costs": Even though the reference you used for setting the initial electrolysis costs is basically comprehensible, they seem rather low compared with other literature (e.g., <https://doi.org/10.1016/j.apenergy.2020.114780>, https://www.irena.org/-/media/Files/IRENA/Agency/Publication/2020/Dec/IRENA_Green_hydrogen_cost_2020.pdf, <https://www.ipa.fraunhofer.de/content/dam/ipa/de/documents/Publikationen/Studien/Studie-IndWEDE.pdf>). While it becomes more reasonable knowing that these values are for capacities of 100MW per system, this should at least be stated. As to my knowledge, there is no system of this size in operation up to now, so it is rather a hypothetical value. Furthermore, as you only focus on alkaline electrolysis (which is also the cheapest so far) as the "general type" in your analysis, taking this rather low (related to all types) starting value, adds some significant bias. Even more so, as this impacts the overall electrolysis cost level in the whole model. Therefore, a proper sensitivity analysis on the impact of the initial system costs is a must in my opinion.

AR: We have added a statement in the main paper about the initial cost assumptions with an additional comment that these cost assumptions are for a 100 MW plant. There was a first boom of electrolysis mainly to produce ammonia fertilisers in the 20th century before the steam methane reforming process became more cost-competitive. Alkaline plants at 100 MW scale have been built from the 1920s in Norway, and later in Canada, Egypt and India [1]. Today, alkaline electrolysis on a MW scale are already operational in many countries (e.g. 150 MW_{el} in Aswan (Egypt) [2], 20 MW_{el} in Fredericia (Denmark), which should be extended by 2025 to 300 MW_{el} and by 2030 to 1 GW_{el} [3]). In China, alkaline electrolysis above > 100 MW_{el} capacity are already operational or planned within the next year, e.g. 150 MW_{el} in Ningxia (operational), 260_{el} MW alkaline electrolysis in Xinjiang (should be completed by mid-2023) [4].

As suggested by the reviewer we run a sensitivity analysis in the Supplementary Material with additional scenarios with higher initial cost assumptions for electrolysis in which we vary the investment costs of electrolysis in 2020 in the range of 650-5850 EUR/kW_{elec} (see also comment above). We find that hydrogen continues to be produced by electrolysis and the investment costs reduce over the modelling horizon. The hydrogen volume and usage is comparable to our base assumptions for investment costs in 2020 in the range of 650-1300 EUR/kW_{elec}. With higher initial costs the production of grey hydrogen increases and the usage of hydrogen is reduced by up to 15% in 2050 with initial cost assumptions of 5850 EUR/kW_{elec}.

RC: Section "Comparing endogenous and exogenous learning": p. 9, 1st para: You should probably recap that technological learning occurs as a function of produced (cumulative) volumes and not as a function of time. Thus, the sequential method is more accurate than the completely exogenous model

(which would still presume cost reductions even without any production).

AR: We modified the previous description of the three methods and added a statement emphasising that the investment costs in the sequential method in contrast to the exogenous method depend on the installed capacities of a technology.

p.10

■ *We compare three different methods of modelling investment cost reduction typically used in ESM, including (i) the endogenous method where investment costs are adjusted according to the installed capacity, (ii) the standard exogenous method with given fixed cost trajectories for each technology and investment year and (iii) the sequential method. In the sequential method, the optimisation problem is first solved using the exogenous cost assumptions, then the investment costs are updated depending on the optimised installed capacities. The same experience curves are assumed as in the endogenous method. The process of solving and updating investment costs is iteratively repeated until the difference of investment costs between two iterations is below a threshold. The threshold is set to a maximal mean square difference between optimised investment costs of the current and the previous iteration of 5%. The sequential method is a linear problem that requires less computational resources than the MILP of the endogenous method, but unlike the exogenous method, the investment costs are adjusted based on the installed capacities.*

RC: p. 11, top: "... the total annualised costs are overestimated by 9% and investments in emerging technologies are delayed." - this is probably dependent on how the threshold for the iterations is set.

AR: The higher costs and delayed investments are caused by the applied method and not the threshold for the iterations. The sequential as opposed to the endogenous method does not have a foresight of how much the investment costs can decrease. For a new technology with high investment costs in 2020 but with a high learning rate, the endogenous method can leverage the possibility of a strong reduction in investment costs through an expansion of the technology. The sequential method does not have this foresight, as the investment costs are adjusted depending on the installed capacities after each optimisation run. The delayed investments are also a result of the method. The endogenous method has the foresight to first invest in a technology so that the investment costs decrease. The sequential method does not have this foresight and therefore delays the investment to a later period. We have reformulated the section comparing the different methods to make the advantages and disadvantages from them clearer.

RC: Section "Limitations of this study":

- the model only considers production-related scaling effects, effects of unit scaling (individual scales of the installed electrolyzers) are neglected

- spill-over effects between electrolysis (see above)

AR: Thank you for the valuable additions to the limitations. We were discussing already spill-over effects but not between electrolysis. We have added them to the discussion section, as well as the limitation that we do not consider unit scaling.

p.15

■ *No spill-over effects between technologies are taken into account. These effects could be important for example for on- and offshore wind, but also for spill-over between different electrolysis types.*

RC: "In all scenarios, the electricity generation from renewables is greater than the electricity demand of the electrolysis at every point in time. It is therefore possible, that all hydrogen produced via electrolysis is green..." - I assume that "at every point in time" narrows down to every calculated year. So this statement is true for "on balance". There may be times, where storage is needed for a continuous production or conventional generation, due to non-correlation supply and demand.

AR: By the expression 'at every point in time' we mean that for **every hour** modelled, the electricity generation from renewables is greater than the electricity demand of electrolysis, **not** averaged over a year. We have now clarified this statement in the revised version.

p. 16

█ *In all scenarios, the electricity generation from renewables is greater than the electricity demand of the electrolysis at every modelled hour.*

2. Reviewer #2

RC: **Page 2: The authors mention that a comparison of 15 IAM found that costs have fallen below 2050 expectations. There is no reference included. Does that mean it was done as part of this work? I follow that those findings highlight the need to avoid setting high floor costs, but that statement alone does not show the need to remove restrictions on expansion rates or max feed-ins. Maybe there is another reason for limiting expansion rates in modeling (subsidy limitations, manufacturing limitations, or building queue limitations). This sentence should be reworded so as not to overstate the findings from the IAM comparison. If the authors want to make the points regarding expansion rates and max feed-ins please include additional references to support that statement.**

AR: The comparison of the 15 IAMs is referring to a study by Krey et al. [5]. We have clarified this in the manuscript and added four additional studies supporting this statement. We have reformulated our statement in the revised version and are now referring to exogenous assumptions that are excessive high or low, such as floor costs or annual expansion rates.

p.2

█ *The investment costs of solar have decreased rapidly due to successful policy support. A comparison by Krey et al. [13] of the cost assumptions in 15 Integrated Assessment Models (IAM) shows that the cost of PV in 2020 have already fallen below the model expectations for 2050. Way et al. [8] show that the progressive cost projections for solar, onshore wind, batteries and polymer electrolyte membrane (PEM) electrolysis from several IAMs and the International Energy Agency (IEA) are high compared to historical developments or even above costs in 2020. Several other studies [14–17] criticise that the link between cost reduction and capacity installation is not well represented in models and that exogenously-set constraints such as floor costs or excessively low annual growth rates lead to an underestimation of cost reductions. This highlights the need to model investment costs endogenously without assuming excessive high floor costs, extremely constraining restrictions on maximum annual expansion rates or maximum penetration of renewable energies.*

RC: "This highlights the need to model investment costs endogenously without assuming too high floor costs, restrictions on expansion rates or maximum feed-ins of renewable energies."

- Introducing constraints on expansion rates or max feed-in do not necessarily change the results. I agree that removing those constraints can improve the model; however, you make it sound like this is a critical problem that needs to be solved. Particularly when we are looking at a trade-off between computational time and additional features or resolution, why do you think this is important? I can see a scenario where the expansion rate and max feed-in are appropriately set so they don't affect the outcome of the optimization and instead the computational capacity is used to increase spatial resolution resulting in potentially more accurate results. Please support why you think this is important to remove these constraints versus improving other features.

- Word choice for "too". Change to "excessive" or something similar

AR: - The impact of the assumed constraint (e.g. annual maximum expansion rate) on the results depends on how close the limitation is to the optimal solution. If the conditions are close to or above the optimal rate of expansion, they do not have a strong influence on the results. In the revised version, we have added reference to numerous other studies showing that energy models have consistently underestimated cost reductions through exogenous assumptions about annual growth rates or floor costs (see section above). This is therefore a critical problem in modelling technology developments since the determination of an upper value is influenced by the presumption of the modeller. For example, a faster scale-up is conceivable through political support or public pressure, as the production of the COVID vaccine has shown. If the limits of the constraint are chosen so high that they are not binding, firstly the reduction of computational effort is lower compared to binding constraints and secondly it is still not ensured that the constraint nevertheless influences the result (e.g. a too tight annual build-out rate of a new technology prevents investment in it, since no cost-optimal investment costs can be achieved with the given limit). In this study, we want to determine the cost-optimal annual expansion rates. In a sensitivity analysis, we then determine the effects if these cost-optimal expansion rates are not achieved. We have added a justification on page 2 after the sentence 'We do not specify the expansion rates of renewable capacities nor CO₂ emission targets for individual years.'

- We changed the word 'too' also in the following sentences to 'extreme' or 'excessive' in the revised version.

RC: **Page 2: "We do not specify the expansion rates of renewable capacities nor CO₂ emission targets for individual years. These additional conditions often lead to a lower computational effort but also predetermine the transition paths."**

- The use of the phrase expansion rate without additional qualification is confusing. In this case I assume you are referring to maximum annual rate but please update to make this clear.

- Secondly, if you are referring to max annual expansion rate, you have not made the case for why this is important. There are practical factors that can limit max annual installation capacities (building permits, subsidies/credits, installation time, etc.). You have not sufficiently justified that free expansion is appropriate.

- As mentioned before, from a modeling perspective if you set the max expansion rate sufficiently high it will not affect the results as the authors have suggested; however, this exception was not mentioned.

- Lastly, despite the discussion in the paper about the importance of relaxing expansion rates, you performed a sensitivity in which build out rates of renewable generation capacity are limited. I am seeing mixed signals in your narrative regarding the most appropriate way to proceed and there is no justification either way. Please justify the relaxation of max annual capacity limits.

AR: - We have clarified the term 'expansion rate' in this sentence and also in the rest of the manuscript. It is now always described as 'maximum annual expansion rate'.

- We added further literature in the introduction showing that cost reductions are underestimated due to exogenous assumptions about maximum annual growth rates or floor costs. We have added a justification in the revised version why we think a free expansion rate is appropriate (see also answer above).

- We added a statement about sufficiently set constraints, which do not influence the results. However, the determination of the limit for these constraint is difficult since non-binding is not a sufficient measure for the limit (see answer above).

- We first want to determine the cost-optimal annual growth rates and then analyse the impact if these capacity expansions are not reached.

RC: **Page 3: "cost and efficiency assumptions of alkaline electrolysis cells (AEC)"**

- While alkaline is the market leader now, PEM is rapidly growing. It is not readily apparent that alkaline will be the market leader in 2050.

- Including detailed learning algorithms is a good way to explore the potential for other competing electrolysis technologies. The authors need to justify the use of only alkaline by discussing PEM costs expectations and discussing how their inclusion might impact the results.

- AR: In the Supplementary Material, we have added a summary of the cost assumptions of different sources and different types of electrolysis and performed a sensitivity analysis on the assumed investment costs (see Section 'Investment cost of electrolysis' in the Supplementary Material). We refer to this analysis in the main manuscript. The investment costs in 2020 are varied between 650-5850 EUR/kW_{elec}. By varying the investment costs, hydrogen continues to be produced by electrolysis and the investment costs reduce over the modelling horizon. With higher initial investment costs in 2020 of 1300 EUR/kW_{elec}, which is in the range of PEM investment costs in 2020, investment cost decrease to 189 EUR/kW_{elec} in 2050. The hydrogen volume and usage is comparable to our base assumptions for investment costs in 2020 in the range of 650-1300 EUR/kW_{elec}. With higher initial costs the production of grey hydrogen increases and the usage of hydrogen is reduced by up to 15% in 2050 with initial cost assumptions of 5850 EUR/kW_{elec}.
- RC: **Page 3: figure 1: Use different line styles (pattern, color and/or thicknesses) for the fully exogenous and mixed lines. It is unnecessarily difficult to distinguish between them.**
- AR: We have adjusted the colour, line thickness as well as the line style in the figure to make it easier to distinguish between the individual categories.
- RC: **Page 4: figure 2: Please change the legend to better define the scenarios in figure 2. 1p5 should be +1.5°C, or something similar. 1p5 is never defined – we are left to infer.**
- AR: The legend in Figure 2 is changed in the revised version to +1.5°C,+1.7°C,+2.0°C.
- RC: **Page 4: figure 2: You mentioned that SCC can drive the total cost for the +1.7C scenario below the +2.0C scenario, but figure 2 caption specifically says (without climate damage included) so please describe why the +1.7C goes below the 2.0C scenario in figure 2. This directly reflects on the credibility of the model and should be discussed.**
- AR: The total annualised costs in Figure 2 are without the social costs of climate damage (SCC) as described in the caption. The +1.7°C scenario leads to lower total costs in 2050 even without climate damage costs, as the tighter budget requires more investments in CO₂ neutral infrastructure in earlier years compared to the +2°C budget. However, as the latter also has to fulfil the condition of CO₂ neutrality in 2050, larger investments with associated higher costs are necessary in 2050, leading to higher total annualised costs in 2050. Still, the summed costs over the entire modelled period are lower for the +2°C budget than for the +1.7°C budget, if no costs for climate damages are taken into account. If costs for climate damages are added, the +1.7°C budget is the most cost-effective (see also Supplementary Material Figure 26). We have added an additional explanation to the section on why the costs of the +1.7°C budget in 2050 are lower than those of the +2°C budget.
- RC: **Page 6: What is included in the investment cost? (stack, balance-of-plant, installation, land, grid connection) Additionally, you need to mention how operating and maintenance costs are included in the main text. I see from the supplemental that a value is provided for electrolyzer but what is included in that value (labor, water, stack replacement, etc.)?**
- AR: We have added a paragraph to the introduction in the main manuscript describing what is included in the investment costs. We further added a section 'Investment costs of electrolysis' in the Supplementary Material with more detailed information and a sensitivity run concerning the initial cost assumptions of electrolysis.

The investment costs of the electrolysis, also called engineering, procurement and construction (EPC) price, include costs of stack, power electronics, balancing of the plant, installation. They do not include costs for grid connection, land, water purification, transformer costs. The operating and maintaining costs are estimated to be 2% in [6] based on current projects. They do not include stack replacement since a replacement during the lifetime of the plant is not necessary, assuming a technical lifetime of AEC stack of more than 100 000 hours and about 4000 full load hours of the electrolysis.

RC: Page 6: The supplemental material includes costs for electricity distribution grid and grid connection, but it is never mentioned how those are used. Are these flat costs added to the investment cost? Do those electricity infrastructure items benefit from learning as well?

AR: The grid connection costs are added to the investment costs for renewable energies, as wind and solar are built decentrally to the distribution grid. The electrolysis is assumed to be connected centralised and no additional grid connection costs are added. The grid connection costs are mentioned in the main manuscript (p.6 in the first submitted version), as stated they do not undergo any technology learning. They are added as flat costs to the investment costs. Electricity distribution grid costs have to be paid for the capacity of the distribution grid, which is the interaction between the transmission level (e.g. generation of wind farms, solar utility but also conventional power plants, large scale storage, electrolysis) and the distribution level (e.g. electrified heating sector, residential electricity demand, battery electric vehicles, home batteries). We added a paragraph to the description of the model which explains how the costs for the distribution grid are applied.

p. 6 (already in version before revision)

■ *One should be aware that grid connection costs are added to these investment costs in the model that do not undergo any learning. For offshore wind grid connection costs depend on the location and connection type, for solar and onshore wind additional grid connection costs of 133 EUR/kW_{elec} are added.*

p. 20 (new in revised version)

■ *We optimise the power transfer capacity between transmission and distribution level. No existing grid infrastructure of distribution grids is assumed. Losses in distribution are neglected. Costs of the distribution grid of 500 EUR/kW_{elec} are applied. Electricity demands, heat pumps, resistive heaters, rooftop PV, home batteries and electric vehicles are connected to the distribution grid. All remaining technologies of the power sector (e.g. large scale storage, wind parks, conventional power plants, electrolysers) are connected to the high-voltage grid.*

RC: Page 6: Readers can benefit from seeing the typical breakdown of hydrogen production by end use. This is not well described in the text but it is presented in the supplemental. Please either add a figure showing a representative example breakdown of end uses for hydrogen or add additional text describing the general behavior and a reference to the supplemental material for more information.

AR: We discuss the hydrogen use and supply of in the manuscript in the section "Installed electrolysis capacities and H₂ usage". We have expanded the description in the revised version and added a reference to the Figure in the Supplementary Material with the breakdown of hydrogen demand and supply.

RC: Page 6: The finding that no blue hydrogen is used is an important finding but only receives a few sentences and an additional sensitivity scenario. The additional scenario performed is one in which SMR+CC benefits from endogenous learning while electrolysis and renewables does not. This is not enough to increase use of blue hydrogen, but that is a really weak scenario. The dynamics of what is happening are not explained. I am left to assume that the cost for SMR+CC is greater than electrolysis+renewables even in the sensitivity, but it is never stated by the author. Please add discussion about the cost dynamics that result in this result. Also, if you never see blue hydrogen, how are you sure it is

working correctly. As a reader, I'm not sure that your model correctly includes blue hydrogen. The sensitivity analysis should have instead looked at the price point at which we start seeing blue hydrogen to provide some context for why it isn't included. Only performing a scenario with endogenous learning added to SMR+CC does not adequately address the subject of blue versus green hydrogen.

AR: In the revised version, we have added a detailed analysis of the conditions under which blue hydrogen is used. The use depends on the investment costs, the CO₂ storage potential and the carbon capture rate. With our base assumptions for CO₂ storage potential (200 Mt_{CO₂}/a) and capture rate (90%), blue hydrogen is used from investment costs below 286 EUR/kW_{H₂}. A large sequestration potential of more than 2000 Mt_{CO₂} per year with base assumptions on investment costs (572 EUR/kW_{H₂}) and capture rate (90%) leads to a production of blue hydrogen with a share of 19% of total production. With optimistic assumptions on capture rate (100%) and CO₂ storage potential (2000 Mt_{CO₂}/a), most of the hydrogen is produced as blue hydrogen from investment costs below 286 EUR/kW_{H₂}.

RC: Page 7: Similar to the question about electrolyzer costs, what is included in the renewable cost? Is this overnight costs? What is included in O&M?

AR: We have added a section in the Supplementary Material describing the composition of renewable generation investment costs and O&M and we are referring in the main manuscript to this section. The investment costs are overnight costs.

Main manuscript p. 4

■ Further details about the initial investment cost assumptions of renewable generation are given in the Supplementary Material.

Supplementary Material p. 20

■ In the following, we provide an overview of the composition of the investment costs for wind and solar. The exogenous cost assumptions are from the Danish Energy Agency technology data DEA [6] which also provides a detailed description of each technology. The technology assumptions depend on the build year of the respective asset. External grid connection costs are added to these investment costs in the model that do not undergo any learning. For offshore wind grid connection costs depend on the location and connection type (AC or DC), for solar and onshore wind additional grid connection costs of 133 Eur /kW are added.

The investment costs of onshore wind consist of the costs for equipment (turbine, foundation, cables), installation and development, cost of land, internal grid connection, decommissioning cost of existing turbines and other costs (for example compensation of neighbours living close to the wind park). The investment cost of offshore wind include cost for equipment (turbine, foundation, cables, grid connection), installation, project development and other costs (e.g. insurances, sea right fees, contingencies). Operating and Maintenance cost of wind farms include insurance, service agreement, repairs not covered by service agreement, land rent and administration. Solar PV investment costs include costs for the equipment (PV module, inverter, transformer, grid connection, balance of the plant), the installation and other costs (e.g. costs for permits, surveys, studies, planning, legal expenditures). Operational and maintenance cost of solar PV include insurance, land rent, cleaning of the modules, asset management and grass cutting.

RC: Page 7: There is some effort to compare the results to previous work (e.g., REPowerEU plan), but this is only done for specific results and not across all of the scenarios (e.g., not done for the +1.5C case). As such, I'm not sure how realistic and reasonable these results are. That in turn, limits the usability and impact of the results. Without a more detailed comparison, this is a nice technical study but not

actionable from a system planning or electrolyzer manufacturer point of view. Please include more comparisons to related work to understand the results in the context of other models. I understand that no other study has introduced learning-by-doing with hydrogen, but there are, in addition to the one that you provided, other studies that look at achieving climate targets or high renewable share scenarios (without hydrogen or a limited representation of hydrogen) that could be used to provide some insight into the magnitude of the renewable capacity needed which could be compared to your results.

AR: We have added a comparison to other studies in the revised version in the revised version. Our finding for cost of green hydrogen production are in good agreement with several other studies [7–9]. Our model offers the advantage of a higher temporal resolution to better reflect system advantages of electrolysis, but unlike the studies mentioned above, we only assume local learning for electrolysis and do not model global trends. Unlike other studies [10–12], we do not assume imports of hydrogen, which would lead to lower installed capacities of electrolysis and renewables. In some studies [8, 13], consistent with our results, green hydrogen production is cheaper and favoured. In other studies [10], blue hydrogen contributes up to 52% of total demand.

RC: Page 8: The "seqcost" scenario is introduced but not described in that section or the supplemental. I finally found out that it was sequential cost method on page 9, but the term "seqcost" is not included in the paper text at all. Please add a description of the "seqcost" scenario near its first use on page 8 and also in the supplemental. Not being able to search for it in the text cost me 10 minutes to figure out what "seqcost" meant.

AR: We have added a description to the manuscript text as well as to the captions of the figures in the manuscript and the Supplemental Material.

RC: Page 9: Please add a description for why "Seqcost" is important. As I understand it, the seqcost scenario essentially allows learning to be applied to the technologies while maintaining linearity. Does that accurately capture the meaning of seqcost? If so, why are the results not more similar to the endogenous scenario? I think the reader could benefit from more information on how the investment cost is changed (because it is clearly a function of capacity so it must be based on the learning rate since endogenous is only based on time and not capacity). I assume that the seqcost utilizes the same learning rate expression and value that the endogenous scenario does, but the author needs to include more details about how the seqcost scenario is implemented? Finally, what threshold was used and is it an important assumption?

AR: We have added a more detailed description of how the sequential cost method is implemented. The threshold is based on the maximum mean square difference of the investment costs of a learning technology between two iterations. The threshold in this study is set to 0.05. A higher threshold requires fewer iterations and thus a shorter solution time, but can lead to fewer convergent solutions.

We further included a paragraph describing why the results of the sequential cost method are different compared to the endogenous method.

p. 10

■ *In the sequential method, the optimisation problem is first solved using the exogenous cost assumptions, then the investment costs are updated depending on the optimised installed capacities. The same experience curves are assumed as in the endogenous method. The process of solving and updating investment costs is iteratively repeated until the difference of investment costs between two iterations is below a threshold. The*

threshold is set to a maximal mean square difference between optimised investment costs of the current and the previous iteration of 0.05. The sequential method is a linear problem that requires less computational resources than the MILP of the endogenous method, but unlike the exogenous method, the investment costs are adjusted based on the installed capacities.

RC: Page 9: Hydrogen costs represent production and storage costs, right? I have not seen any discussion on the distribution/delivery costs. Please describe how those costs are considered in your work. There is one region so am I left to assume that there is no cost for hydrogen distribution and delivery? Further, hydrogen storage costs are included in the supplemental material of table 2 but it is never discussed how the hydrogen storage is sized. Please describe how hydrogen storage is sized.

AR: Delivery costs of hydrogen are not considered in the main results. We have added a paragraph stating this. The uses of hydrogen considered in this model for synthetic fuels, industry and heavy-duty transport are relatively centralised. The option to use hydrogen for heating in households is not used. For delivery a hydrogen transmission system would be necessary but not a distribution grid as they exist for electricity and gas today. In the revised version we included a sensitivity scenario in the Supplementary Material with a higher spatial resolution which considers costs for a hydrogen grid. These cost have no large impact on the total system costs with a share of 0.1-2% of total annualised system cost. We added a description of the size of the hydrogen storage to the main manuscript.

p. 8

■ *Hydrogen is stored in salt caverns and the energy capacity ranges between 278-349 TWh from the +2.0°C to the +1.5°C scenario. These storage capacities are comparable to existing natural gas storage of 1075 TWh, once adjusted for the lower volumetric energy density of hydrogen, and below the European technical potential of 84.8 PWh_{H₂} [35]. No costs are assumed in the main results for the distribution of the hydrogen. A sensitivity analysis of a scenario with a higher spatial resolution and corresponding costs for electricity and hydrogen infrastructure, as well as an annual breakdown of hydrogen supply and usage and installed capacities is shown in the Supplementary Material. In scenarios with a higher spatial resolution, the hydrogen and electricity grids account for a share of 0.1-7.6% of the total system costs, higher electrolysis capacities are installed and total system cost increase by 12-16%. The trade-offs of an electricity and hydrogen grid in a decarbonised European energy system are discussed in more detail in a further publication [36].*

RC: Page 9: The "Comparing endogenous and exogenous learning" should be reworked to focus more on a comparison of the three methods. At present it provides some anecdotal explanation of the results based on the selected exogenous curve. Assuming a low learning rate would change the magnitude of electrolysis installed, the cost of hydrogen and the total system cost, so the authors should speak more generally about the endogenous versus exogenous results than on the baseline learning rate results. Overall, I think the paper is missing discussion about the merits of the different methods (endo., exog., and seqcost) and that is what I would like to see included once this section is reworked.

AR: We have restructured the section to focus on a comparison of all three methods and clarify the advantages and disadvantages of each method.

RC: Page 10: why aren't there bars for seqcost in 2020 and 2025? Is there no difference or were those left out? Please print the values for each bar on the chart to make it clear.

AR: There is no difference in total system costs for 2020 and 2025. We have included in the revised versions the numbers for each bar in the figure.

RC: Page 10: How can figure 6 be an absolute value, while the other figures include plus or minus 10%.

I am left to assume that the difference uses the base endogenous scenario. Please add the results for plus and minus 10% learning rate for the endogenous scenarios to the figure. Also the authors have identified that the +1.5C is difficult to reach so why is it the only comparison point between methods. I need to see the results for +1.7C and +2.0C to be able to discern the importance of implementing an endogenous learning strategy.

AR: The figure shows the percentage cost difference of the sequential and exogenous method compared to the endogenous method with base learning rate. We have added for all budgets figures in the Supplementary Material that also show scenarios with varied learning rates. Further we have added the graphs for the +1.7°C and +2.0°C scenarios to the Supplementary Material (see section ‘Annualised total system cost difference between the methods’).

RC: Page 10: Similarly, I think figure 6 is one of the main outcomes of this work but it is not well described or put in context with the tradeoffs between cost, technology delay, computational benefit and spatial resolution. Only one instance is shown for endogenous versus seqcost (21hrs versus 1hr, with a 9% overestimation when compared to the base endogenous scenario with +1.5C.

- First, based on the values in figure 6 what does the 9% value in the text represent? No values are greater than 7% in figure 6.

- Second the authors need to include a more comprehensive discussion about the tradeoffs between methods. It appears that in 2050 the energy balances between the three methods are quite similar (fig 10-13). For the +1.5C case, if the energy balances are quite similar in 2050 (fig 13e) and the

- You can't relate them since the costs per unit are not comparable

AR: -We must apologise, the value was wrong. Thank you for reading the manuscript in such detail. We have removed it from the text since we also restructured this section, the maximum deviation is 7% for +1.5°C and 13% for the +2.0°C scenario.

- we have reformulated this section (see comment above) to discuss the trade-offs between all three methods. The energy balance in 2050 is mainly impacted by the assumed carbon budget and not the applied method. Nevertheless, there are differences in the produced hydrogen volume. E.g., in scenarios corresponding to a +1.7°C budget 8% and 6% less hydrogen is produced in the sequential and exogenous method in 2050. In the endogenous method, hydrogen is more used for methanation and for re-electrification in fuel cells compared to the other two methods.

- It is true that the investment cost per unit are different between the methods. But also the installed capacities differ, for example the exogenous method leads to lowest installed electrolysis capacities in 2050 followed by the sequential method with -33% and -17% compared to the endogenous method in the +1.5°C scenarios.

RC: Page 11: How are FOM costs affected by learning? More systems operating is likely to materially affect maintenance costs. I assume like efficiency and lifetime, learning is not applied to FOM. If that is right, please include in limitations.

AR: FOM costs are not subject to endogenous learning but are exogenously assumed depending on the technology and the year of construction. We have added this to the limitations of the study.

p. 15

■ *We only model technology learning for investment costs, but not for fixed operational and maintenance cost (FOM), efficiencies or lifetimes, which are assumed exogenously based on the year of construction and the respective technology.*

RC: Page 11: What can you say about the relative impact of the limitations? What should be the highest priority targets items to include in the next version of the model (e.g., 1) endogenous efficiency, 2)

endogenous lifetime, 3) greater spatial resolution, ...)?

AR: We have revised the section on limitations of the study and now categorise the relative influences of individual limitations on the results according to our personal judgement. We also list possible future work.

RC: **Page 11: Regarding spatial resolution, I am particularly interested in the impact of assuming that there is only one region in Europe. Please provide a sense for the impacts on system cost for this assumption. I think this is important because lots of effort is put into including endogenous cost estimates, but if the spatial simplification greatly underestimates costs, then it might be better to introduce seqcost method with multiple regions in Europe. I recognize that this may be beyond the scope of the analysis; however, when the authors make such strong statements about why it is important to include endogenous cost estimates, then it is necessary to support their claims by considering the tradeoffs between alternative strategies. (also see spatial resolution comment for page 14)**

AR: We have added a sensitivity analysis regarding spatial resolution to the Supplementary Material. Here we evaluate the influence of spatial resolution on the exogenous method. A higher resolution leads to higher installed capacities of electrolysis. For the endogenous or sequential method, this would lead to lower investment costs of electrolysis. The investment cost of electrolysis described in the manuscript thus represent an upper limit with regard to spatial resolution but a lower limit in terms of total system costs. Since the sequential method is less computationally expensive than the endogenous method, it would be possible to apply it with higher spatial resolution. Of course, this still comes with the disadvantage of the method that investments are delayed and that it has no foresight of possible cost reductions.

RC: **Page 12: At several points in the paper (including page 12), the author shows that exogenous cost estimation methods lead to higher costs, but this result is somewhat anecdotal and a function of the learning rate selected compared to the exogenous cost estimation. A low learning rate could be worse than the exogenous case, and an optimistic exogenous cost curve could be lower system cost than the baseline endogenous estimate. To that end, the discussion should focus more on the strengths and weaknesses of each of the methods and not focus on their resulting values. Implementing an endogenous method for assessing costs is not inherently better, but it represents a tradeoff between different aspects. This will require that you rework the "Comparing endogenous and exogenous learning" section the "limitations" section and the "conclusion" section to focus more on tradeoffs.**

- For example, the fourth paragraph of the conclusion discusses that "ignoring the virtuous cycle... leads to severe distortion of the results"; however, this paper only shows one example of this based on a range of learning rates and one set of exogenous prices from DEA and does not provide sufficient verification of these results.

- The last paragraph of the conclusion is not something that you have proven from your work. The first sentence is speculation and not specific enough to be verified. The other two sentences are not adequately proven by this paper. There was a lot of modeling work that was done, but the scenario development, sensitivities and discussion in this paper do not work together to verify the hypothesis in the last paragraph of the conclusions. I personally understand the value of integrating endogenous cost estimates into these models but you have not made a clear and strong case for that.

AR: We thank the reviewer for their careful consideration of our conclusions. We have reformulated the section comparing different methods and the conclusion. We focus the discussion in the revised version more on the trade-offs between the different methods. Further, we have concretised our statements in the conclusion.

RC: **Page 14: "...no maximum build out rate of renewable generation capacity and no annual CO2 emission paths are specified but also predetermine the transition paths"**

- See similar comment from text on page 2. Please reword to clarify.

AR: We have reworded the expression to clarify that our statement refers to maximum annual expansion rates of renewable generation capacity.

RC: Page 14: "To make the optimisation problem still computationally solvable, it is aggregated spatially and temporally" a single node in Europe

- It sounds like you are trading a max buildout rate constraint for severely limited spatial resolution. This renders energy transmission infrastructure immaterial. Distribution and transmission infrastructure is an important feature for the cost and reliability of electricity, gas, and hydrogen networks. I see that you include some variability for renewables (6 profiles), but a key unanswered question is how the transmission networks evolve to meet future production and demand of electricity, natural gas and hydrogen.

- Please justify your simplification of the model to one node for Europe.

AR: Even with the constraint of annual maximum build-out rates, we need to simplify the model in terms of spatial and temporal resolution to make it solvable. We have reworded the sentences to avoid giving the impression that the omission of maximum annual expansion rates is balanced against spatial resolution. We have added a sensitivity analysis to the Supplementary Material, in which we compare scenarios with a higher spatial resolution with the exogenous method to our simplified optimisation problem with a lower spatial resolution. Infrastructure costs only contribute to a small part of the total costs. Electrolysis capacities increase with higher spatial resolution. Therefore, our investment costs with low spatial resolution can be seen as an upper estimate. Total system costs increase by 20% with a higher spatial resolution and can be seen therefore as a lower estimate. The minor share of infrastructure costs in the total costs is also shown in another publication by Neumann et al. [14]. This publication focuses on the trade-off of the electricity and hydrogen grid in a decarbonised European energy system.

RC: Supplemental

1. Several figures in the supplemental material are too small to read. Please increase the figure size and text size to make all figures legible. - Figures 10-13 need unique and detailed captions.
2. Figure 11 f is incorrectly repeated from fig 11 d
3. The table starting on page 16 does not have a caption. Also, there are no units on that table so I can't tell if FOM values are absolute or percentage of capital cost. Please add units for every row in the table.
4. I mentioned in an earlier comment but it is not clear what is included in the "investment" cost. Since you didn't say capital cost, I assume it is capital and installation at least. Please add a paragraphs of text before the table describing some of the details of what is included in the investment cost.
5. Please add references for where each cost item comes from in the second table.
6. Specifically for electrolysis costs, FOM is 2 (€/kW-year, I presume), which when compared with the investment cost of €650/kW in 2020, over a 25 year timeperiod it appears that no stack replacements will be performed or at least they aren't part of the FOM. Readers need the reference to be able to understand what is included. Because of the importance to the results, please justify the values selected for electrolysis and compare with other sources.

7. In the second table not all technologies have a lifetime. Why not, and if no lifetime is included, then what is assumed?
8. Because of all of the legend entries, Figures 10-13 are not particularly useful. You can keep the figures in the supplemental, but I recommend inserting the values in table form as well. Readers will be able to explore this data more easily than trying to compare across figures. Also, you should include installed power in addition to a table that shows energy.
9. The supplemental has a good discussion section for each figure until the "energy balance" section, after which, there is no discussion for any of the subsequent figures. Given the importance of figures 10, 13, 15-16, the reader would benefit from some discussion for those figures.

AR: Supplementary Material

1. In the revised version of the Supplementary Material we increased the font size of previous Figures 10-13 and added a unique caption to them.
2. The Figures 11f and 11d in the previous version were different but the caption was incorrectly repeated. In the revised version we have corrected this.
3. We added a caption and units for every row of the table with the technology cost assumptions.
4. The investment cost include cost for equipment and cost for installation. We have added a paragraph in the revised version which is describing what is included in the investment costs.
5. We added a reference for every row of the table with the technology assumptions.
6. The FOM of the electrolysis is 2% of investment costs in 2020, which means 13 Eur/kW-year. We have added the reference to the table. We added in the revised version of the Supplementary Material a discussion and a sensitivity analysis of the investment cost assumptions of electrolysis. Stack replacement is not included in the investment costs. The lifetime of current AEC stacks is more than 100000 hours, assuming the electrolysis is running 4000 hour per year the stacks do not need to be replaced during the lifetime of the electrolysis [6].
7. We have restructured the table with the technology assumptions in the revised version. If no lifetime is given, a lifetime of 25 years is assumed. This is also described in a paragraph before the table.
8. We have included the corresponding tables to the energy balances figures. Further we have included tables for all budgets displaying the installed capacity of each technology.
9. We have added a description to the energy balances for different carriers in the revised version.

3. Reviewer #3

RC: I would suggest to find more references agreeing with or opposing the points made by Way et al. [8], so it would support your result or not.

AR: We have adjusted the statement after corresponding with the authors of the paper. In fact, our statements do not contradict the study by Way et al. The revised version of the manuscript now contains the following

statement:

p. 15

■ *Our CO₂-unconstrained scenario is less expensive than scenarios with a CO₂ budget. The study by Way et al. [8] shows that a scenario that follows historical trends and does not allow the transformation of comparatively expensive sectors is more costly than a scenario with a +1.5°C budget. However, unlike Way's scenario, in the CO₂-unconstrained scenario parts of the energy system can transform if it is cost-effective.*

RC: It seems that 1.7degreesC is achievable more so than 1.5degreesC and less emissions than 2degreesC, could this be a useful recommendation from this study?

AR: The necessary capacity expansion of renewables and electrolysis is easier to achieve in the +1.7°C budget than in the +1.5°C budget. However, an increase of more than 1.5°C budget leads to more far-reaching environmental impacts, such as the extinction of most coral reefs [15]. We therefore do not want to present the 1.7°C scenario in this paper as a compromise solution between necessary annual expansion rates and additional emissions. If we want to avoid far-reaching impacts on the environment, it is necessary to adhere to the 1.5°C budget. It should also be noted that we assume no sufficiency measures, no negative emissions and no endogenous transformation of the transport sector, which could lead to lower expansion rates and make the 1.5°C scenario easier to achieve. It is also not clear that the 1.7°C scenario is the most cost-effective option if social cost of carbon is included (depending on its assumed value). We discussed this in a previous publication [16].

RC: We suggest the authors to check the numbers in this section: “between 171-237 e/kW for solar PV, 818-900 e/kW for onshore wind and 1327 e/kW - grid connection costs are added to these investment costs in the model that do not undergo any learning”

AR: These costs refer to investment costs (without grid connection) in 2050 with the endogenous method and base learning rate assumptions. We have checked the values and they agree with our results and the referenced graph. We have reworded the sentence to make it clear that the values refer to the base learning rates and the endogenous method.

RC: We suggest to edit the text as there is a double link added in ref 27.

AR: Thank you for pointing this out. We have removed the duplicate link from the source.

RC: Regarding supplementary data Table 6 on page 16, 17 & 18 is not labelled and you only have reference 5 as the sole source of data; which is from 2019. Is this prudent? Should multiple sources be used to align expected costs and technology?

AR: We have labelled all the tables in the revised version and added units and further sources to the table with the technology assumptions. We have adopted most of the technology assumptions from the Danish Energy Agency [6], which includes many technologies, as we firstly want to use consistent data and thereby avoid picking out overly optimistic or pessimistic assumptions for individual technologies. Second, the data is constantly updated (last update was in April 2022, we have updated the year of the source in the revised version). In order to investigate uncertainties in the investment costs of electrolysis, we have added a sensitivity analysis in the revised Supplementary Material.

RC: Also add a number of acronyms to the list e.g. FOM & VOM

AR: We have added the acronyms FOM and VOM to the list. Further we included a definition of the two below

the technology assumptions tables.

References

- [1] Marcelo Carmo, NEL Hydrogen, **Introduction to Liquid Alkaline Electrolysis** (2022).
URL <https://www.energy.gov/sites/default/files/2022-02/2-Intro-Liquid%20Alkaline%20Workshop.pdf>
- [2] B. Mathiesen, I. Ridjan, D. Connolly, M. Nielsen, P. Vang Hendriksen, M. Bjerg Mogensen, S. Højgaard Jensen, S. Dalgaard Ebbesen, Technology data for high temperature solid oxide electrolyser cells, alkali and PEM electrolysers, Department of Development and Planning, Aalborg University, 2013.
- [3] **Everfuel is leading the flagship project, HySynergy, in establishing a 20 MW PtX facility.** , accessed on 3.11.22 (2022).
URL <https://www.everfuel.com/projects/hysynergy/>
- [4] Leigh Collins, **Record breaker - World's largest green hydrogen project, with 150MW electrolyser, brought on line in China** , accessed on 3.11.22 (2022).
URL <https://www.rechargenews.com/energy-transition/record-breaker-world-s-largest-green-hydrogen-project-with-150mw-electrolyser-brought-on-line-in-china-2-1-1160799>
- [5] V. Krey, F. Guo, P. Kolp, W. Zhou, R. Schaeffer, A. Awasthy, C. Bertram, H.-S. de Boer, P. Fragkos, S. Fujimori, C. He, G. Iyer, K. Keramidas, A. C. Köberle, K. Oshiro, L. A. Reis, B. Shoai-Tehrani, S. Vishwanathan, P. Capros, L. Drouet, J. E. Edmonds, A. Garg, D. E. Gernaat, K. Jiang, M. Kannavou, A. Kitous, E. Kriegler, G. Luderer, R. Mathur, M. Muratori, F. Sano, D. P. van Vuuren, **Looking under the hood: A comparison of techno-economic assumptions across national and global integrated assessment models**, Energy 172 (2019) 1254–1267. doi:10.1016/j.energy.2018.12.131.
URL <https://doi.org/10.1016/j.energy.2018.12.131>
- [6] Danish Energy Agency (DEA), **Technology Data** (2022).
URL <https://ens.dk/en/our-services/projections-and-models/technology-data>
- [7] IRENA, **Green Hydrogen Cost Reduction: Scaling up Electrolysers to Meet the 1.5°C Climate Goal**, Abu Dhabi (2020).
URL https://www.irena.org/-/media/Files/IRENA/Agency/Publication/2020/Dec/IRENA_Green_hydrogen_cost_2020.pdf
- [8] Energy Transitions Commission, **Making the Hydrogen Economy Possible: Accelerating Clean Hydrogen in an Electrified Economy** (2021).
URL <https://www.energy-transitions.org/publications/making-clean-hydrogen-possible/#download-form>
- [9] International Energy Agency (IEA), **Global Hydrogen Review 2022** (2022).
URL <https://iea.blob.core.windows.net/assets/c5bc75b1-9e4d-460d-9056-6e8e626a11c4/GlobalHydrogenReview2022.pdf>

- [10] G. S. Seck, E. Hache, J. Sabathier, F. Guedes, G. A. Reigstad, J. Straus, O. Wolfgang, J. A. Ouassou, M. Askeland, I. Hjorth, H. I. Skjelbred, L. E. Andersson, S. Douguet, M. Villavicencio, J. Trüby, J. Brauer, C. Cabot, **Hydrogen and the decarbonization of the energy system in europe in 2050: A detailed model-based analysis**, Renewable and Sustainable Energy Reviews 167 (2022) 112779. doi:10.1016/j.rser.2022.112779.
URL <https://doi.org/10.1016/j.rser.2022.112779>
- [11] International Renewable Energy Agency, **Global hydrogen trade to meet the 1.5°C climate goal: Part III – Green hydrogen cost and potential** (2022).
URL <https://www.irena.org/publications/2022/May/Global-hydrogen-trade-Cost>
- [12] R. Ortiz Cebolla, F. Dolci, E. Weidner Ronnefeld, **Assessment of hydrogen delivery options** (KJ-NA-31-199-EN-N (online)) (2022). doi:10.2760/869085.
URL <https://publications.jrc.ec.europa.eu/repository/handle/JRC130442>
- [13] BloombergNEF, **Hydrogen Economy Outlook** (2020).
URL <https://data.bloomberglp.com/professional/sites/24/BNEF-Hydrogen-Economy-Outlook-Key-Messages-30-Mar-2020.pdf>
- [14] F. Neumann, E. Zeyen, M. Victoria, T. Brown, **Benefits of a hydrogen network in europe** (2022). doi:10.48550/arXiv.2207.05816.
URL <https://doi.org/10.48550/arXiv.2207.05816>
- [15] N. Wunderling, J. F. Donges, J. Kurths, R. Winkelmann, **Interacting tipping elements increase risk of climate domino effects under global warming**, Earth System Dynamics 12 (2) (2021) 601–619. doi:10.5194/esd-12-601-2021.
URL <https://doi.org/10.5194/esd-12-601-2021>
- [16] M. Victoria, E. Zeyen, T. Brown, **Speed of technological transformations required in Europe to achieve different climate goals** (2022). doi:10.1016/j.joule.2022.04.016.
URL <https://doi.org/10.1016/j.joule.2022.04.016>

REVIEWERS' COMMENTS

Reviewer #1 (Remarks to the Author):

The authors improved their article based on the given review comments. All my previous comments were addressed and result in a comprehensive analysis of endogenous learning for electrolysis in a European energy model under different climate targets. However, in my second review I came across another potential issue which I think is needed to be addressed (or clarified) before publication.

- Is the value given in Table 1 for "Global capacity" of electrolysis used as your initial value for cumulative capacity in the experience curve model? If so, other studies (e.g., <https://doi.org/10.1016/j.ijhydene.2008.03.011>, <https://doi.org/10.1038/nenergy.2017.110>, <https://doi.org/10.1016/j.ijhydene.2019.09.230>, <https://doi.org/10.1016/j.apenergy.2020.114780>) on experience curves for electrolysis also consider past installations of chlor-alkali electrolysis, which would rather result in a cumulative capacity of >21 GW. This would probably have a significant impact on your results, especially since you clearly focus on AEC and also justified your presumption of large-scale plants with these early installations. Please clarify.

- Comparison to other results:

As you already point out that differences in hydrogen costs may be attributed to different full load hours and electricity costs, it would probably be better to (additionally) compare CAPEX directly.

Reviewer #2 (Remarks to the Author):

The authors have implemented extensive revisions and addressed all of my previous concerns.

Please find below the point by point responses to the reviewer comments. The differences compared with the previous submitted version of the paper are highlighted in blue and red text in an attached file.

1. Reviewer #1

RC: Is the value given in Table 1 for “Global capacity” of electrolysis used as your initial value for cumulative capacity in the experience curve model? If so, other studies (e.g., <https://doi.org/10.1016/j.ijhydene.2008.03.011>, <https://doi.org/10.1038/nenergy.2017.110>, <https://doi.org/10.1016/j.ijhydene.2019.09.230>, <https://doi.org/10.1016/j.apenergy.2020.115848>) on experience curves for electrolysis also consider past installations of chlor-alkali electrolysis, which would rather result in a cumulative capacity of >21 GW. This would probably have a significant impact on your results, especially since you clearly focus on AEC and also justified your presumption of large-scale plants with these early installations. Please clarify.

AR: Yes, the value in Table 1 for "Global capacity" is used as the initial value for the cumulative capacity in the experience curve.

We only consider alkaline water electrolysis capacity here and do not include chlor-alkali electrolysis. This approach follows other literature [3]. Chlor-alkali electrolysis and alkaline electrolysis are different processes with distinct purposes. Chlor-alkali electrolysis primarily produces chlorine and caustic soda, with hydrogen as a byproduct. AEC electrolysis is designed specifically for hydrogen production. Comparing these two technologies directly is not entirely accurate, as they serve different purposes and have different optimisation targets. There are other factors, like different market dynamics (hydrogen production versus chlor-alkali production) and different policy and regulatory frameworks, which can impact the scaling of these technologies. Thus, we did not incorporate the cumulative capacities of chlor-alkali electrolysis when assessing the learning curve of alkaline electrolysis.

According to IEA [1], global alkaline electrolysis should be roughly 1 GW between 2021-2022. In the latest version (10/10/2022) of the hydrogen project database [2] which includes already decommissioned electrolysis, cumulative capacities sum up to 630 MW. This data was also used in another recent publication by Odenweller et al. [3]. Conversely, one could also argue for lower cumulative capacities, as merely 250 MW of alkaline electrolysis has been installed between 2000-2019 [4], that there is technological forgetting/unlearning, and lower cumulative capacities should be assumed.

In the SI, we only refer to current already operational and planned projects of electrolysis [5, 6] to justify the assumed investment costs for a 100 MW plant and not to chlor-alkali electrolysis plants. We clarify now in the table of the revised version that we only consider alkaline electrolysis for the cumulative capacity. We further move the table to the corresponding section of the paper. We added your comment about the impact on the

learning curve and investment costs in case that the initial cumulative capacity the capacities of chlor-alkali electrolysis is included to the limitations section.

p. 6 footnote in table for the cumulative capacity

■ *only considering alkaline electrolysis and no other electrolysis type.*

p. 5 in the SI

■ *The assumed current global cumulative capacity for the experience curves significantly influences the results. For example, if the global capacity of electrolysis is assumed to include the capacities of chlor-alkali electrolysis, the cost reductions through technology learning would be significantly lower. If, on the other hand, one assumes the capacities of PEM or SOEC, which are well below 1 GW today, the investment costs would decrease more.*

RC: **Comparison to other results:** As you already point out that differences in hydrogen costs may be attributed to different full load hours and electricity costs, it would probably be better to (additionally) compare CAPEX directly.

AR: This study focuses on the costs of green hydrogen production, not the investment costs of electrolysis. We have therefore chosen the costs of hydrogen production for comparison with other studies. The costs for hydrogen production depend on several factors, such as the costs of electricity generation from renewable sources, costs for electrolysis or the full load hours of electrolysis. We optimise endogenously not only the investment costs of the electrolysis but also the operation of the electrolysis and the investment costs of the renewables. A comparison based only on the investment costs of electrolysis thus only shows a partial aspect of the costs for green hydrogen production considered in this study.

Due to the editor's request, the sections "Comparison to other studies" and "Limitations" are moved to the SI and briefly summarised in the Discussion section. The main paper focuses on the overall costs of green hydrogen production due to the above arguments. However, we have now added a comparison of the investment costs of electrolysis to the discussion in the SI.

p. 4 in the SI

■ *The electrolysis investment costs in our findings are below the projected investment costs in most other studies. For example, the highest investment costs in our scenarios with a +2°C budget are 380 €/kW_{elec} in 2030, while Fraunhofer ISE finds 444 €/kW_{elec} [8] and IEA 400 €/kW_{elec} [9]. The costs for producing green hydrogen are similar compared to other studies since we have lower optimal full load hours. In 2050, we find investment costs of electrolysis between 75-95 €/kW_{elec}, which is well below the assumptions of IEA and IRENA of 200 €/kW_{elec} [9, 10]. Vartiainen et al. [4] find investment costs in the lower range of our findings of 80 €/kW_{elec}.*

References

[1] International Energy Agency, **Electrolysers** (2022).

URL <https://www.iea.org/reports/electrolysers>

[2] International Energy Agency (IEA), **Hydrogen Projects Database** (2022).

URL <https://www.iea.org/data-and-statistics/data-product/>

hydrogen-projects-database

- [3] A. Odenweller, F. Ueckerdt, G. F. Nemet, M. Jensterle, G. Luderer, **Probabilistic feasibility space of scaling up green hydrogen supply**, Nature Energy 7 (9) (2022) 854–865. doi:10.1038/s41560-022-01097-4.
URL <https://doi.org/10.1038/s41560-022-01097-4>
- [4] Danish Energy Agency (DEA), **Technology Data Renewable fuels**, Version number: 0009, accessed 18.4.2023 (2022).
URL https://ens.dk/sites/ens.dk/files/Analyser/technology_data_for_renewable_fuels.pdf
- [5] Leigh Collins, **Record breaker - World's largest green hydrogen project, with 150MW electrolyser, brought on line in China**, accessed on 3.11.22 (2022).
URL <https://www.rechargenews.com/energy-transition/record-breaker-world-s-largest-green-hydrogen-project-with-150mw-electrolyser-brought-on-line-in-china-2-1-1160799>
- [6] **Everfuel is leading the flagship project, HySynergy, in establishing a 20 MW PtX facility.**, accessed on 3.11.22 (2022).
URL <https://www.everfuel.com/projects/hysynergy/>